# NEO-STIM advances personalized neoantigen-specific adoptive T cell therapy

Divya Lenkala[1], Jessica Kohler[1,5], Brian McCarthy[1], Michael Nelson[1,6], Noor A. M. Bakker [2], Renate de Boer[2,7], Emily K. Jackson [1], Joong Hyuk F. Sheen[1,8], Susan Hannes[1], Ekaterina Esaulova[1], Kai Stewart[1], Claudia Gottstein [3], John Attanasio[1,9], Flavian D. Brown[1,10], Sebastian Hymson [1,11], Shirisha Meda[1], Maaike van Zon[2], Saskia Scheij[2], Rhianne Voogd[2], Brenda Raud [2,12], Ziyan Xu[1], Jessica S. W. Borgers [2,13], Maartje W. Rohaan[2], Kristen N. Balogh[1], Asaf Poran [1], Michael Rooney[1], Jesse Z. Dong[1], John R. Srouji[1], Vikram R. Juneja[1], Christina M. Arieta[1], Cynthia M. Nijenhuis[2], Bastiaan Nuijen[2], Mark DeMario[1], Kelledy Manson[1], Ton N. M. Schumacher [2,4], Richard B. Gaynor[1], John B. Haanen [2], Joost H. van den Berg[2,14] & Marit M. van Buuren [1] ✉

Neoantigen-based adoptive T cell therapies (ACTs) represent a promising avenue in cancer immunotherapy due to their exquisite tumor specificity. The first cell-based immunotherapy for a solid tumor, comprising tumor-infiltrating lymphocytes, recently received FDA approval. Building on this, we designed a distinct ACT approach, where T cell responses against personalized neoantigens are systematically generated from autologous peripheral blood. Here we report the establishment of NEO-STIM, an ex vivo induction process to prime and expand pre-existing memory and de novo CD8[+] and CD4[+] T cell responses, thereby highlighting critical parameters for generating potent neoantigen-specific T cell responses. The drug products comprise mutant-reactive, polyfunctional, and cytotoxic CD8[+] and CD4[+] T cells, able to recognize autologous tumor material. Following infusion, T cell responses are detected in tumor and blood of a patient, and display activated/exhausted and cytotoxic phenotypes. A first-in-human clinical trial (NCT04625205) recently further validated proof-of-concept, supporting continued development of this ACT approach.

Cancer immunotherapy has become an established treatment modality in oncology. Adoptive T cell therapies (ACTs) in particular have advanced rapidly, resulting in recent FDA approval of a tumor-infiltrating lymphocyte (TIL) product defining a major milestone for personalized adoptive cell therapy[1]. While early ACT approaches targeted primarily tumor-associated antigens (TAAs) on liquid tumors[2,3], more recently the focus has shifted to solid tumors and targeting neoantigens[4–8]. Neoantigens are peptides derived from mutant

[1]BioNTech US, Cambridge, MA, USA. [2]Netherlands Cancer Institute, Amsterdam, The Netherlands. [3]BioNTech SE, Mainz, Germany. [4]Oncode Institute, Utrecht, The Netherlands. [5]Present address: Clasp Therapeutics, Cambridge, MA, USA. [6]Present address: Johnson & Johnson, Springhouse, PA, USA. [7]Present address: AstraZeneca, Amsterdam, The Netherlands. [8]Present address: Repertoire Immune Medicines, Cambridge, MA, USA. [9]Present address: Yale University, New Haven, CT, USA. [10]Present address: Rhapsogen, Inc., Cambridge, MA, USA. [11]Present address: University of Pittsburgh; School of Medicine, Pittsburgh, PA, USA. [12]Present address: Sanquin, Amsterdam, The Netherlands. [13]Present address: University of Colorado; Anschutz Medical Campus, Denver, CO, USA. [14]Present address: Galapagos NV, Leiden, The Netherlands. ✉e-mail: Marit.vanBuuren@biontech.us

proteins generated as a consequence of genomic mutations in tumor cells. They are unique for each patient and thus require a personalized approach. Neoantigen recognition has been shown to be a major component in TIL therapy, where TILs are obtained from the tumor, expanded ex vivo and then re-infused into the patient following a lymphodepleting regimen.

Some of the key challenges in TIL therapy are i) only a relatively low number of tumor neoantigens (0.5–2.3%) being generally recognized by T cells[9–14], ii) the necessity for a sizeable and resectable amount of tumor tissue, and iii) the relatively high proportion of phenotypically exhausted T cells used for TIL (compared to blood-derived T cells)[15].

To address these challenges, we designed a neoantigen-specific T cell therapy and established NEO-STIM, an ex vivo T cell induction platform to STIMulate T cells from peripheral blood to generate responses against tumor NEOantigens. We envisioned a streamlined, scalable process that would generate autologous T cell drug products (DP) with the following characteristics: i) inducing polyclonal de novo responses (see below for definition), in addition to expanding T cells from the memory compartment, with the objective to broaden the diversity of the responses and reduce the risk of immune evasion[16,17]; ii) containing T cell responses of both CD8+ and CD4+ subtype, capable of direct cytotoxicity or production of cytokines upon challenge with neoantigen-expressing cells[9,18,19]; iii) containing T cell responses with high specificity for the mutant epitope to minimize off-target side effects.

NEO-STIM builds on a previously reported vaccination approach where clinical benefit was seen in patients treated with personalized neoantigen-based peptides in combination with immune checkpoint blockade[20]. It takes that approach one step further: Instead of vaccinating in vivo, T cell responses against personalized neoantigens are induced ex vivo under tailored and optimized conditions to avoid immunosuppressive mechanisms by removing the cells from the inhibitory niches within the tumor microenvironment (TME) seen in cancer patients[21].

Here, we report the successful design, optimization, and scale-up of NEO-STIM, first using model antigens in healthy donors (HDs), then using peptides representing personalized neoantigens from patients with melanoma. We describe critical parameters for the generation of potent T cell responses and for robust and scalable manufacturing. Phenotypic characterization of T cells from the DP and from tumor and blood of a patient pre- and post-infusion provides insights into the biology of the T cell response in the context of this ACT approach.

## Results

### Development of NEO-STIM to generate neoantigen-specific T cells

The overarching goal was to develop a personalized, autologous ACT DP from peripheral blood (Fig. 1A). To facilitate platform development, NEO-STIM was initially carried out using HD peripheral blood mononuclear cells (PBMCs) and model antigens that either always (MART-1), frequently or sporadically (HIV-derived epitopes, and previously

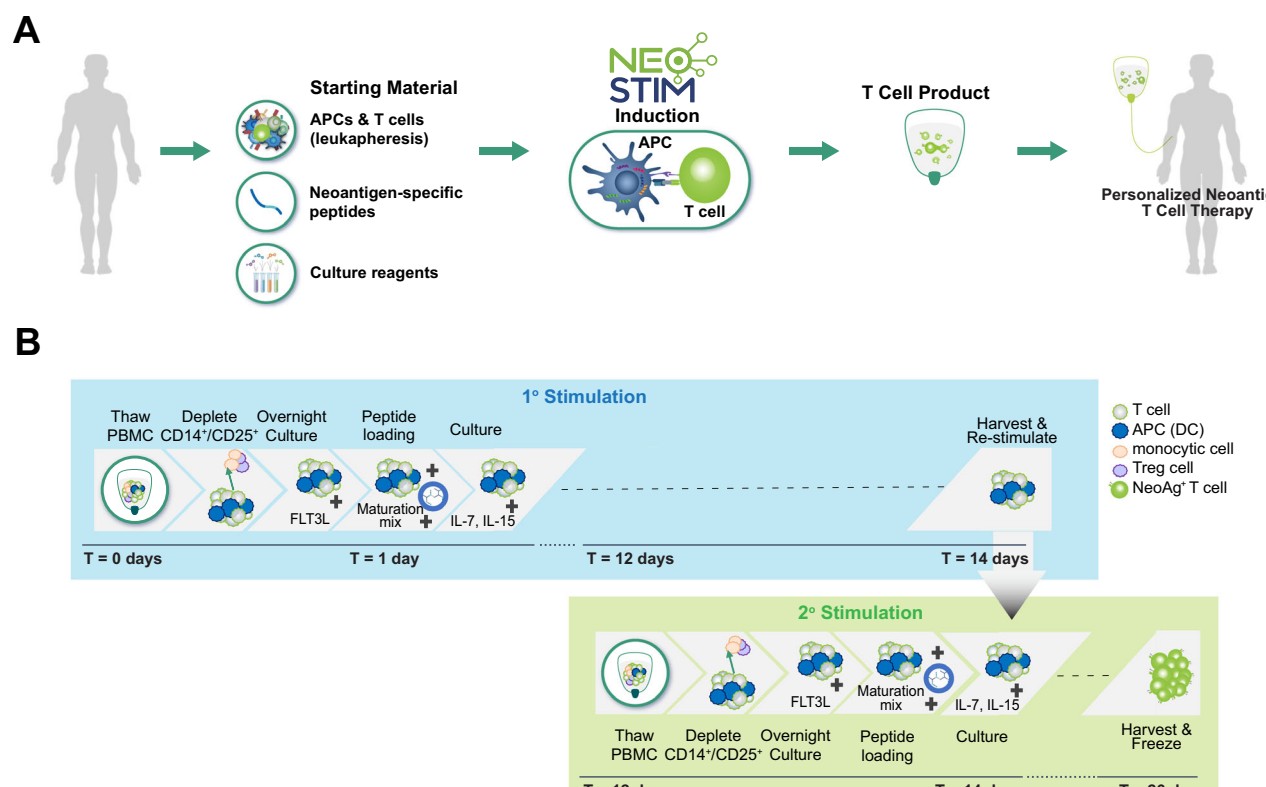

**Fig. 1 | Needle-to-needle process to generate a personalized neoantigen-specific T cell product. A** Overall approach, starting from the leukapheresis to obtain peripheral blood mononuclear cells (PBMCs) from the patient to the infusion of the final drug product (needle-to-needle process). Neoantigen-specific peptides are generated for each patient (Starting Material) and used to prime and expand the patient's T cells (Induction), before infusing the T cell product into the patient. **B** Detailed overview of the NEO-STIM process with staggered cultures for initial (1°) stimulation and restimulation (2° stimulation). Optimized parameters were APC type, culture time for 1° culture, initiation time point for 2° culture, cell ratios between primary and subsequent cultures, total culture time and number of stimulations. This resulted in the final NEO-STIM protocol: For the 1° stimulation, PBMCs were depleted of CD14+ and CD25+ cells. 25-50 million depleted cells, containing CD11c+ naturally circulating dendritic cells were cultured overnight in the presence of FMS-like tyrosine kinase 3-ligand (FLT3L), then peptide loaded, matured with maturation cocktail (TNF-α, interleukin (IL)−1β, Prostaglandin-E 1 and IL-7), and cultured for 14 days with IL-7 and IL-15. A 2° culture from a new aliquot was initiated on day 12. Depleted cells were cultured overnight, primed, matured and combined with the 1° culture on day 14 followed by expansion for an additional 12 days. On day 26, the cells were harvested and frozen until analysis. Created in BioRender. Gottstein, C. (https://BioRender.com/sxm5zeh).

identified neoantigens) were able to prime T cell responses in vitro[22–24] (see Supplementary Table 1). The size of the TCR repertoire and epitope characteristics are key aspects that impact the generation of a T cell response. As such, dividing model antigens into both highly and weakly immunogenic categories allowed us to systematically analyze their distinct response dynamics and growth kinetics, ensuring experimental decisions were tailored to the unique characteristics of each epitope class. Model antigens were matched with HD material expressing relevant human leukocyte antigen (HLA) restriction elements to investigate the expansion of memory-like responses (MART-1) and the induction of responses from the naïve compartment (neoantigens and HIV-derived)[22–24]. Finally, T cell products were generated from patient-derived PBMCs and their relative potency was measured against engineered and natural tumor targets.

To optimize cell type composition for T cell induction, NEO-STIM is initiated with the isolation of PBMCs and depletion of inhibitory cell subsets. The immunogens are then added, the antigen presenting cells (APCs) are matured, and finally antigen-specific T cells are primed and expanded (1° culture). Unstimulated PBMCs are thawed, and a secondary (2°) culture is prepared and stimulated analogous to the 1° culture in a staggered way, and parallel cell cultures are combined and expanded to a final DP. The restimulation of primary cultures with autologous depleted PBMCs aims to further expand already primed antigen-specific T cell responses and prime additional T cell responses with the objective of producing a drug product that is enriched for multiple neoantigen-specific T cell responses. Figure 1B illustrates the steps of the final process, and the following critical process parameters were optimized to finalize this: choice of APCs (see next paragraph), starting cell number, culturing time, and aspects of the restimulation process (cell number and ratio, timing, and overall number of restimulations). The respective optimization experiments and results are described in detail in Supplementary Figs. 1 and 2. When optimizing the timing between 1° and 2° culture, we observed that the time difference between starting the 2° culture and restimulation had an impact on the balance between new priming events and expansion of already primed events, in addition to the frequency of neoantigen-specific cells. In general, decisions to implement process steps into NEO-STIM were driven both by biology (phenotype, number, and magnitude of induced responses) and by feasibility of good manufacturing practices (GMP) manufacturing (logistics, costs, and time). The final optimized protocol is described in the caption of Fig. 1B, and yielded T cell responses against the memory-like antigen MART-1 as well as de novo responses against high and low immunogenic model viral and neoantigens (Supplementary Fig. 1E, 2D, E). De novo responses were defined as responses that were not detected in the pre-NEO-STIM sample. This definition is practical for the purpose of this study, as responses could either originate from T cells in the naïve compartment or from very small memory clones that were below the detection limit of the assay. However, here the responses are most likely derived from the naïve T cell compartment, given that HD-derived T cells were primed with antigens that the donor has not encountered previously.

### Naturally occurring dendritic cells drive T cell priming in NEO-STIM

With the aim of streamlining T cell priming, we set out to systematically investigate the starting PBMCs to pinpoint critical cell subsets. T cells and APCs are key cell types required for efficient induction of T cell responses from PBMCs. However, cell subsets that potentially prohibit efficient T cell priming, such as monocytic myeloid-derived suppressor cells (MDSCs) and T regulatory (Treg) cells, are also present[25–27]. Therefore, CD25+ T cells, of which the majority were Treg cells, and CD14+ monocytes, for which the short priming period was likely insufficient to differentiate them to mature dendritic cells (DCs), were depleted from the PBMC starting material.

Moreover, since the antigen presentation step is critical for effective T cell priming, we wanted to understand which APC population supports the NEO-STIM process best. DCs are generally considered professional APCs. At the high level, there are three sources of DCs from peripheral blood: classical/conventional DCs (cDCs), plasmacytoid DCs (pDCs) and monocyte-derived DCs (moDCs), with moDCs frequently being used in the context of the manufacturing of TAA-targeted T cell therapy[2,3,28–31]. We compared the priming efficiency of these traditionally used moDCs with that of naturally circulating DCs (nDCs; Supplementary Fig. 3A). Fold expansion, hit rate, frequency, diversity and functionality of CD8+ responses, and additionally the ability to generate CD8+ responses to long peptides (typically used to induce CD4+ responses) were evaluated. nDCs, both cDCs 1 and 2, showed upregulation of maturation markers (CD40, CD80, CD83 and CD86, Supplementary Fig. 3B) and most parameters were more favorable using the nDCs (Supplementary Fig. 3C–G). Notably, CD8+ responses to long peptides were only seen with nDCs.

To further characterize T cell priming efficiency between cDCs and plasmacytoid DCs (pDCs), we depleted either CD11b+ cells or CD11c+ cells or both, in addition to CD14+ and CD25+ cells. cDCs are CD11c+ and CD11b negative to low positive (depending on subtype) while pDCs are double-negative[32,33].

Efficient depletion (90-98.5%) for both markers was achieved in both single- and double-depleted conditions (Supplementary Fig. 4A–D). Depletion of CD11c+ DCs resulted in a significant decline in the total number of antigen-specific cells or hit rate for memory-like MART-1-specific responses and de novo model antigen responses, respectively (Supplementary Fig. 4E, F). This confirms that the CD11c+ cell fraction (primarily cDCs) is critical for T cell priming in NEO-STIM. CD11b depletion had more variable results, consistent with the variability of CD11b expression among the cDC subtypes. pDCs may still be contributing to NEO-STIM, but are not the key stand-alone drivers, based on these results.

These interesting biological findings had practical implications and we conclude that: i) CD14-depletion does not remove any critical DC populations; ii) there is no strong rationale for an additional CD11b depletion step, which would increase the costs of manufacturing; and most importantly iii) nDCs are able to more efficiently prime T cells in NEO-STIM compared to moDCs, which implies that a separate manufacturing workstream of isolating CD14+ cells and maturing them to DCs is neither needed nor beneficial. These conclusions positively impacted workload, time, and costs of NEO-STIM manufacturing.

### NEO-STIM with patient-derived PBMCs efficiently generates responses against personalized neoantigens

To evaluate whether NEO-STIM could efficiently generate T cell responses against personalized tumor antigens, we applied the optimized process to PBMCs from patients with melanoma or ovarian cancer (OVC) indications with a mid-to-high mutational load[34]. This required additional steps, i.e. identifying neoantigens in each patient and then manufacturing the selected personalized peptides to be used as immunogens. Neoantigens were identified and selected by sequencing the patient-specific mutanome and performing epitope predictions in the context of the patients' HLA alleles (both major histocompatibility complex (MHC) class I and MHC class II). The bioinformatic approach considers RNA expression, predicted binding affinity to patient-specific MHC molecules, and predicted proteasomal processing to generate a score and rank the predicted neoantigens[20,35,36]. Additionally, PT03, PT07, PT08, PT09 each had 1-2 pools of previously identified neoantigens[20,22–24,37,38] added to NEO-STIM process.

Research-scale experiments were performed using PBMCs from seven patients, four with melanoma and three with ovarian cancer (see Supplementary Data S1, Supplementary Table 2). After identification of personalized neoantigens, up to 60 synthetic neoantigen peptides per

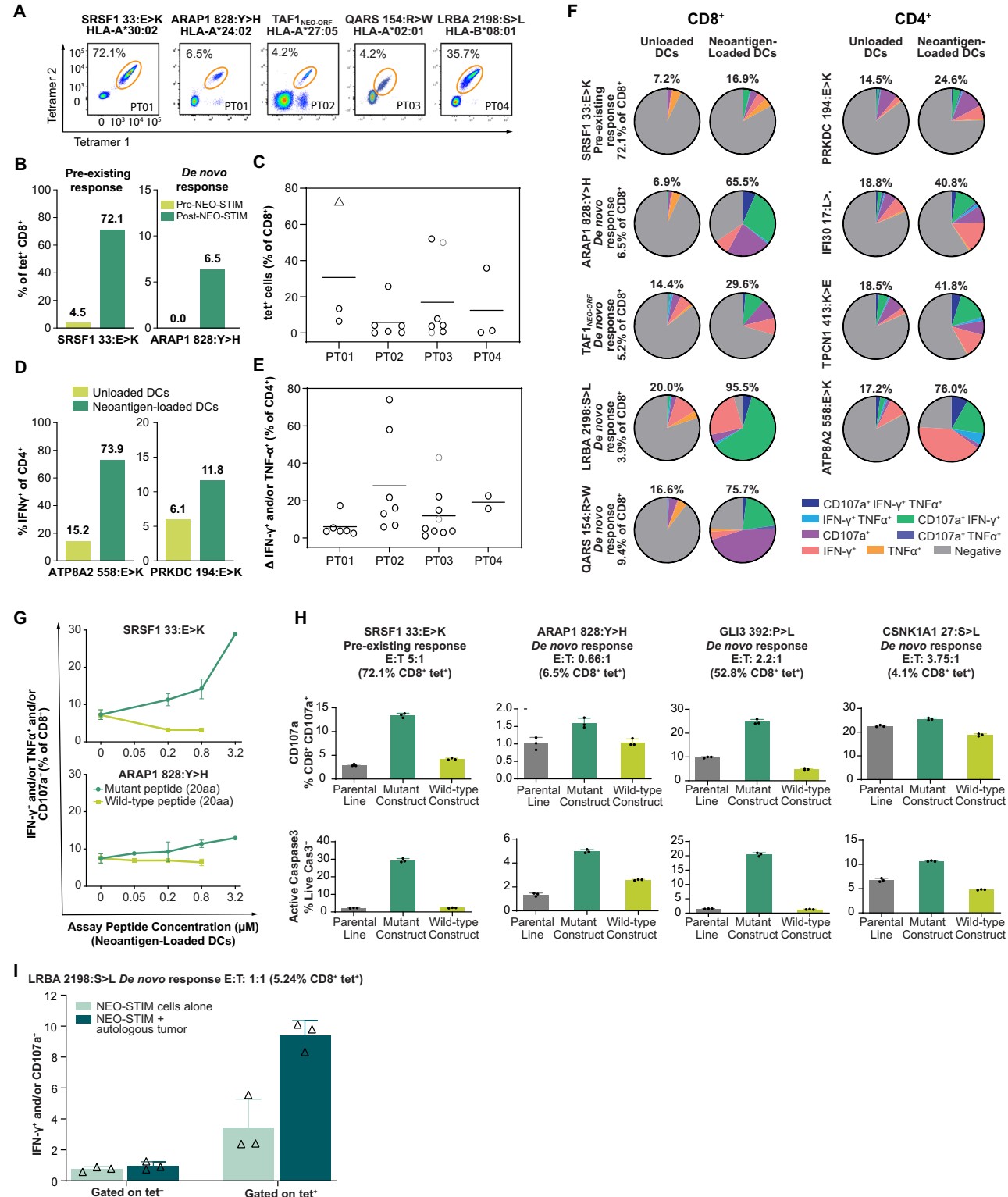

patient were manufactured and divided into six or seven pools. Each peptide pool was used in triplicate (melanoma) or quadruplicate (OVC) to assess reproducibility. After applying NEO-STIM, we investigated both CD8$^+$ and CD4$^+$ responses in the DPs of all seven patients.

To understand which epitopes were immunogenic, we first measured CD8$^+$ responses using pMHC tetramers in a combinatorial coding strategy (Fig. 2A, and Supplementary Fig. 5, 6A). Figure 2A shows five examples of CD8$^+$ responses. One of the CD8$^+$ responses

was detected prior to NEO-STIM (SRSF1$_{33:E>K}$, 4.5% of CD8$^+$ cells) showing that pre-existing responses from the memory compartment can be further expanded. This pre-existing response also showed the highest magnitude in a single well, 72.1% of CD8$^+$ cells (range 51.3–72.1% across three wells), which represented a 16-fold expansion compared to pre-NEO-STIM cells (Fig. 2B). Overall, all seven patients generated multiple CD8$^+$ responses (median 6, range (2–13)) and the magnitude of neoantigen-specific CD8$^+$ cells ranged from 0.04% to 72.1% with a

**Fig. 2 | NEO-STIM process applied to material from patients with melanoma at research scale.** Data from PT01-04, using the optimized protocol (Fig. 1B), except that for PT01-03 three stimulations were carried out (Supplementary Data S1, tab Summary of Responses). **A** Representative flow cytometry plots from last culture day, see Supplementary Fig. 5 for all responses. **B** Example bar graphs showing frequency of tet$^+$ CD8$^+$ T cells of a pre-existing (SRSF1$_{33:E>K}$, left) and a de novo (ARAP1$_{828:Y>H}$, right) response, pre- and post-NEO-STIM (from one replicate of PT01). Tet$^+$ cells: T cells detected by peptide-MHC tetramers. **C** Frequencies of neoantigen-specific CD8$^+$ T cells across all four patients (one out of three replicates plotted). See Supplemental Data S1 tab Research Scale for number of responses tested. Data points represent frequency of an induced response post-NEO-STIM. Triangle symbol: pre-existing response, light gray data points: responses to model antigens. Horizontal lines show mean frequency. **D** Example bar graphs showing frequency of IFNγ$^+$ cells for induced CD4$^+$ cells co-cultured with mutant (MT) neoantigen-loaded dendritic cells (DCs) versus unloaded DCs in post-NEO-STIM samples (from one replicate each from PT02 or PT04, respectively). **E** Frequency of neoantigen-specific CD4$^+$ T cells across all four patient samples. Data points represent the delta of IFNγ and/or TNFα (as a percent of CD4$^+$ T cells) for each response when rechallenged with MT neoantigen-loaded DCs versus unloaded DCs in post-NEO-STIM samples (one out of three replicates plotted). See Supplemental Data S1 tab Research Scale for number of responses tested. Horizontal lines show mean frequency. **F–I** Only responses with sufficient sample availability and magnitude of response were tested. **F** Representative examples of 27 melanoma responses tested. Pie charts depict the polyfunctionality of the identified neoantigen-specific CD8$^+$ T cells (gated on tet$^+$ CD8$^+$ cells, left two columns) and CD4$^+$ T cells (gated on IFNγ$^+$ and/or TNFα$^+$ CD4$^+$ cells, right two columns) upon rechallenge with MT neoantigen-loaded DCs versus unloaded DCs. Percent numbers above pie charts indicate the percentage of functional cells with one, two or three functions. Percent numbers on y axis represent frequency of the response as measured within this assay. $n = 1$-3 technical replicates. **G** Example graphs for MT-specific responses post-NEO-STIM from PT01 (see Supplementary Fig. 7 for all samples) showing IFN-γ$^+$ and/or TNFα$^+$ and/or CD107a$^+$ secretion of total CD8$^+$ T cells. Data shown as mean with SD ($n = 1$-3 technical replicates). **H** Cytotoxicity of T cell responses against tumor cells (parental A375 line): untransduced or transduced with MT or wild-type neoantigen-containing construct and the patient-specific HLA allele. Top panels: Mobilization of CD107a on CD8$^+$ T cells; bottom panels: active caspase 3 on tumor cells. Percent numbers in brackets represent frequency of the response as measured within this assay. Data shown as mean with SD ($n = 3$ technical replicates). **I** LRBA$_{2198:S>L}$-specific T cells from day 14 (STIM 1) sample induced in patient PT04 were co-cultured with a single-cell suspension from the patient's tumor to evaluate recognition through a recall response assay measured by IFNγ$^+$ and/or CD107a$^+$ in CD8$^+$ tet$^+$ and tet$^-$ T cells. Data shown as mean with SD ($n = 3$ technical replicates). Source data are provided as a Source Data file.

median of 4% (Fig. 2C, Supplementary Fig. 5A, 6B, and Supplementary Data S1). Reproducibility between cultures was consistent across biological replicates, in particular for higher frequency responses (Supplementary Data S1).

Next, CD4$^+$ responses were identified by measuring interferon (IFN)γ and/or tumor necrosis factor (TNF)α in response to autologous peptide-loaded vs unloaded DC exposure (Fig. 2D). A functional readout was used with the goal to identify clinically relevant CD4$^+$ T cell responses. All, but one patient, generated multiple CD4$^+$ responses (median 6, range 1-13, Fig. 2E, and Supplementary Fig. 6C, D). The magnitude of neoantigen-specific CD4$^+$ cells ranged from 1.1-74%, with a median of 7.3%) (Supplementary Data S1).

Taken together, NEO-STIM generated a polyclonal T cell product containing multiple neoantigen-specific CD8$^+$ and CD4$^+$ T cell responses using PBMCs from patients with advanced melanoma and ovarian cancer.

## NEO-STIM responses are polyfunctional and recognize mutant epitopes

Prior studies with ACTs showed that a polyfunctional T cell response, i.e. the ability of a T cell to produce multiple cytokines, is correlated with improved clinical outcome[39,40]. We therefore measured the expression of IFNγ, TNFα, and CD107a mobilization in a recall response assay upon stimulation of T cells with their cognate antigen.

Twenty out of 27 CD4$^+$ and CD8$^+$ melanoma responses tested showed an increase in all three functions (IFNγ, TNFα, and CD107a). Overall functionality, i.e. increase in any of the three functions or any combination thereof was detected in 100% of evaluable CD8$^+$ and CD4$^+$ responses (examples shown in Fig. 2F). Polyfunctionality was also observed in one additional ovarian carcinoma patient tested (Supplementary Fig. 6E–G).

Given that neoantigen epitopes can differ from their wild-type (WT) counterparts by as little as a single amino acid, and single point mutations can either reduce or enhance the likelihood of the epitope to be processed and presented[41], we evaluated whether the induced T cell responses could differentiate between mutant (MT) and WT peptides. DCs loaded with varying concentrations of either peptide were co-cultured with the induced T cells. Subsequently, the fraction of responding T cells was quantified based on functional marker expression in response to both peptides. Both of the two evaluable CD8$^+$ responses (SRSF1$_{33:E>K}$ and ARAP1$_{828:Y>H}$) were mutant-specific, which we defined as having a statistically significant reactivity over the no-peptide control (datapoint at concentration 0) without any significant WT reactivity (Fig. 2G, Source Data file, tab Fig. 2G). The one CD4$^+$ T cell response tested at this point of protocol development (TPCN1$_{413:K>E}$) was WT cross-reactive: significant reactivity to MT and WT with no significant increase of MT over WT reactivity (Supplementary Fig. 7B, yellow header, Source Data file, tab Suppl. Fig. 7a).

## NEO-STIM responses are cytotoxic and recognize/respond to autologous tumor

The assessment of cytotoxicity towards antigen-expressing targets is a relevant proxy for in vivo killing capacity. To understand the ability of NEO-STIM-induced CD8$^+$ T cells to kill targets with naturally processed and presented epitopes, we transduced cells of the human melanoma cell line A375 with the associated HLA type and either the MT or WT epitope with flanking natural sequence, positioning the mutation towards the center of the sequence. We then co-cultured the transduced target A375 cells with effector CD8$^+$ T cells from the DP and measured upregulation of CD107a on T cells, which is a surrogate marker for cell killing[42]. In addition, upregulation of active caspase 3, a marker of early apoptosis, was measured on the A375 tumor cells.

Figure 2H illustrates that in four cytotoxic responses (out of five evaluable responses tested), significant upregulation of both CD107a and caspase 3 was observed, when comparing MT-transduced and WT-transduced cells. This showed that the majority of NEO-STIM-induced T cell responses were able to specifically kill mutant antigen-expressing targets.

Finally, the key test for functionality of cytotoxic CD8$^+$ T cells is the ability to recognize and respond to autologous tumor. We were able to obtain a single-cell suspension from a tumor of one patient (PT04). The induced T cells from this patient contained a sizeable CD8$^+$ T cell response towards LRBA$_{2198:S>L}$ after a single stimulation (5.2% tet$^+$ of CD8$^+$ T cells). After co-culture of these T cells with the autologous tumor we measured IFNγ expression and CD107a mobilization, and observed that the LRBA$_{2198:S>L}$-specific CD8$^+$ T cells were able to recognize autologous tumor (9.4% IFNγ$^+$ and/or CD107a$^+$ of tet$^+$ CD8$^+$ cells, Fig. 2I). This demonstrates that NEO-STIM-induced CD8$^+$ T cells can recognize cells derived from autologous tumor and can respond to physiologically relevant levels of presented epitope.

## Polyclonal neoantigen-specific responses show an increased transcriptional activation signature

To assess the phenotypes of NEO-STIM-induced T cells, single-cell RNA/TCR sequencing analysis was performed for a patient (PT01) at four time points: pre-NEO-STIM (day 0), stimulation 1 (STIM1, day 14),

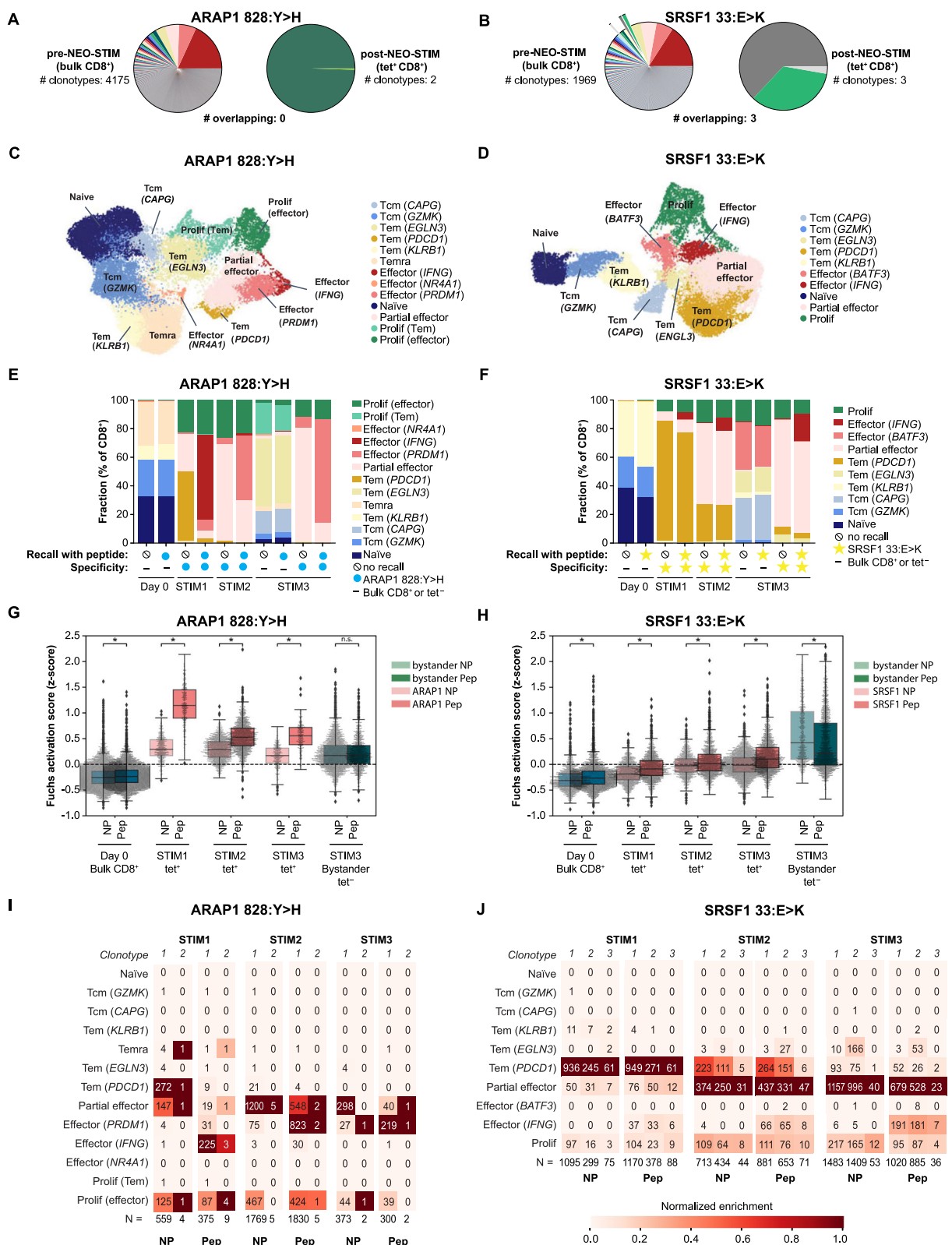

stimulation 2 (STIM2, day 21), and stimulation 3 (STIM3, day 28). PT01 had three CD8⁺ T cell responses in the DP, one of them pre-existing and two de novo, based on tetramer staining (Fig. 2C). We chose the pre-existing (SRSF1$_{33:E>K}$) and one of the de novo responses (ARAP1$_{828:Y>H}$) for this analysis. Single-cell TCR sequencing confirmed that SRSF1$_{33:E>K}$ was indeed a pre-existing response and that ARAP1$_{828:Y>H}$ was a de novo response (Fig. 3A, B). Functional avidity data were generated for

four TCRs (SRSF1: $n = 3$; ARAP1: $n = 1$) and showed mutant-specific binding in the nano- to micromolar range (Supplementary Fig. S8).

At day 0, the percentage of neoantigen-specific cells was extremely low, so the bulk T cells at day 0 were considered by and large as bystanders. For the other timepoints we sorted the cells based on tetramer staining prior to phenotyping. Cells from each time point were co-cultured overnight with DCs loaded with relevant peptide (Pep) or

**Fig. 3 | Transcriptional analysis of NEO-STIM-induced responses ARAP1$_{828:Y>H}$ and SRSF1$_{33:E>K}$ (PT01). A, B** Diversity of the TCR repertoire of samples pre- and post-NEO-STIM. Pre-NEO-STIM: TCR diversity in the bulk CD8$^+$ T cells; post-NEO-STIM: clonality of the tetramer$^+$-sorted CD8$^+$ T cells; # overlapping: number of pre-existing clones. **C, D** Uniform manifold approximation and projections (UMAPs) of multimodal single-cell data of T cell response. Phenotypes were assigned based on known expression signatures, see Supplementary Fig. 9C–F. **E, F** Range of phenotypes across cultures from day 0, 14 (STIM1), 21 (STIM2) and 28 (STIM3), with and without specific peptide challenge. Day 0 cells: sorted on-bulk CD8$^+$, STIM1 and STIM2 cells: sorted on neoantigen-specific (tet$^+$ CD8$^+$), and STIM3 cells: sorted on neoantigen-specific and nonspecific (tet$^-$ CD8$^+$) cells. Tem subsets are characterized by expression of *IL7R*, effector subsets by expression of *TNFRSF9*, and partial effectors by low levels of *TNFRSF9* combined with exhaustion markers (see Supplementary Fig. 9). Note the larger phenotypic shifts in response to antigen for the ARAP1 de novo clonotypes than the SRSF1 memory clonotypes, especially at the first stimulation (STIM1) timepoint. **G, H** Fuchs activation scores based on a published signature[44] for cultures across timepoints (see **E, F**) with and without specific peptide challenge. Boxplots show data quartiles. Outliers were defined using 1.5 times the interquartile range. NP: no peptide; Pep: stimulated with cognate peptide. Statistical analysis was performed using two-sided t-test. ns, $P > 0.05$; *$P \leq 0.05$. Numbers of peptides and exact $P$ values are reported in the Source Data file. **I, J** Heatmap showing phenotypes of cells from ARAP1$_{828:Y>H}$ ($n = 2$ clonotypes, **I**) and SRSF1$_{33:E>K}$ ($n = 3$ clonotypes, **J**) responses across timepoints (see **E, F**) with and without specific peptide challenge. NP no peptide; Pep: stimulated with peptide. Numbers in heatmap: number of cells observed in each phenotype, total number of cells per condition is shown in the line on the bottom. Heatmap colors: scaled relative to the most abundant phenotype for each clonotype. Source data are provided as a Source Data file.

without peptide (NP). Sequences from all conditions were analyzed together and uniform manifold approximation and projections (UMAP) analysis revealed the transcriptional differences between different experimental groups (Fig. 3C, D and Supplementary Fig. 9A, B). Phenotypes were assigned based on expression of known markers (Supplementary Fig. 9C-F).

Upon peptide challenge, ARAP1$_{828:Y>H}$ neoantigen-specific cells shifted drastically, to comprise more effector type cells. Cells with effector phenotypes were of the *IFNG* type at STIM1, but then expressed increasing levels of *PRDM1*, a gene involved in T cell differentiation[43], at STIM2 and STIM3 (Fig. 3E, F). In contrast, phenotypic shifts in SRSF1$_{33:E>K}$-specific T cells upon peptide challenge were detectable, but less pronounced and consisted mainly in the appearance of a small *IFNG$^+$* effector cluster. Overall, the expression of genes such as *IFNG*, *CCL3*, and *TNFRSF9* in cells rechallenged with antigen suggests that these T cells induced through NEO-STIM were functional and have effector function upon rechallenge.

To understand the activation profile of the neoantigen-specific cells upon rechallenge, we used a published T cell activation signature (Fuchs activation score)[44] to compare cells rechallenged with and without cognate peptide. We observed a significant increase in activation scores following peptide stimulation, which was more pronounced in ARAP1$_{828:Y>H}$-specific T cells, particularly at STIM1 (Fig. 3G, H). At the clonotype level, we observed that within both responses all clonotypes had a homogenous expression profile showing an increased fraction of T cells shifting to a "partial effector" population (*IFNG$^{med}$TNFRSF9$^-$* effector cells) with time in culture (Fig. 3I, J), in line with what was observed at the response level.

In summary, two different patterns of phenotypic shift were seen for ARAP1$_{828:Y>H}$ (de novo) and SRSF1$_{33:E>K}$ (pre-existing). The former demonstrated a dramatic shift to effector phenotypes already in STIM1, and from there a gradual shift to a more differentiated effector phenotype (*PRDM1*) in STIM2 and −3 (Fig. 3E), while the latter showed a gradual and less pronounced increase of partial effector and effector phenotypes (Fig. 3F). We analyzed stem-like (*CCR7*, *SELL*) and exhaustion markers (*LAG3*, *PDCD1*, *PRDM1*) in both responses (Supplementary Fig. 9G). *CCR7* and *SELL* showed minimal expression at all timepoints, while *LAG3* and *PRDM1* increased over time with stable *PDCD1* levels. This pattern was consistent across SRSF1 and ARAP1 T cell responses. Considering that *PRDM1* has been shown to be negatively correlated with persistence in CAR-T cells[43] and the fact that the benefits of STIM3 for the pre-existing response (SRSF1$_{133:E>K}$) were minor, only one restimulation was adopted for the final protocol to keep the overall manufacturing time as short as possible.

## Therapeutic-scale manufacturing process generates reproducible results

To support further development for clinical studies, it was critical to scale up NEO-STIM from a small-scale research process to a semi-closed therapeutic-scale process using GMP-grade reagents. The therapeutic-scale process followed the optimized protocol with two stimulations (except for one pilot run, which used three stimulations). Peptides were again divided into at most six distinct pools to avoid overgrowth of dominant clones.

DPs were successfully manufactured across two HDs and three patients with melanoma (Supplementary Data S1 and Supplementary Table 2). Total cell yields ranged from $3.6 \times 10^9$ to $10.2 \times 10^9$ cells and had a high viability ($\geq 79\%$) (Fig. 4A). The majority of cells in all the manufactured DPs were CD3$^+$ T cells ($\geq 60\%$), with the remaining cells being mostly B cells and NK cells (Supplementary Fig. 10A). Additionally, all DPs passed the quality control testing (sterility, mycoplasma, and endotoxin testing). We observed a strong correlation of initial CD8$^+$ frequency with hit rate ($R^2 = 0.87$ for HD and $R^2 = 0.98$ for patient samples) consistent with a similar finding in initial process establishment (Supplementary Fig. 1B), as well as consistency between research scale and therapeutic-scale samples, when comparable processes were used (Fig. 4B). This further highlights the importance of depleting CD14$^+$ monocytes and CD25$^+$ regulatory T cells on day 0 to optimize neoantigen-specific T cell priming.

## Therapeutic-scale DPs are polyclonal, polyfunctional, specific and cytotoxic

Next, we sought to confirm the research-scale results at therapeutic scale. After scale-up, we observed that NEO-STIM can generate multiple T cell responses. Interestingly, variations in growth kinetics across vessels and stimulation with related peptides (e.g., 9mer and 10mer from the same mutation) can lead to non-uniform dilution and, in some cases, unexpectedly varying frequencies of antigen-specific T cells in the final DPs, as observed for TENM3$_{1243:S > L}$ in PT05 (Fig. 4C, Supplementary Fig. 5B and Supplementary Data S1). The majority of cells displayed an effector memory phenotype (Supplementary Fig. 10B) and all showed to be polyfunctional (Supplementary Fig. 10C for examples). Specificity for MT epitopes was tested for 16 therapeutic-scale responses. MT-specificity and WT-cross-reactivity was defined above (section "NEO-STIM responses are polyfunctional"); mutant-selective responses were defined as responses with significant reactivity towards MT and WT peptide, but with a significant increase for mutant over wildtype peptide (together with MT-specific responses labeled as mutant-reactive). Representative plots for each category are shown in Fig. 4D. All of the tested CD8$^+$ responses ($n = 3$) and 85% of the tested CD4$^+$ responses ($n = 13$) were mutant-reactive (Supplementary Fig. 7; Source Data file, tabs Suppl Fig. 7a, b). Given the infrequent observation of WT-reactive responses during development, the release specifications of the drug product did not include the exclusion of WT-reactive clones, thus resulting in the infusion of bulk products[45]. Additionally, all tested neoantigen-specific CD8$^+$ responses were cytotoxic against antigen-expressing tumor lines (Supplementary Fig. 10D, E).

As we showed recognition of autologous tumor at research scale (Fig. 2I), we tested two additional responses (based on sample availability) from a therapeutic-scale DP (ITPR3$_{616:E>K}$ and TENM3$_{1243:S<L}$) with and without addition of cognate peptide. Tumor-

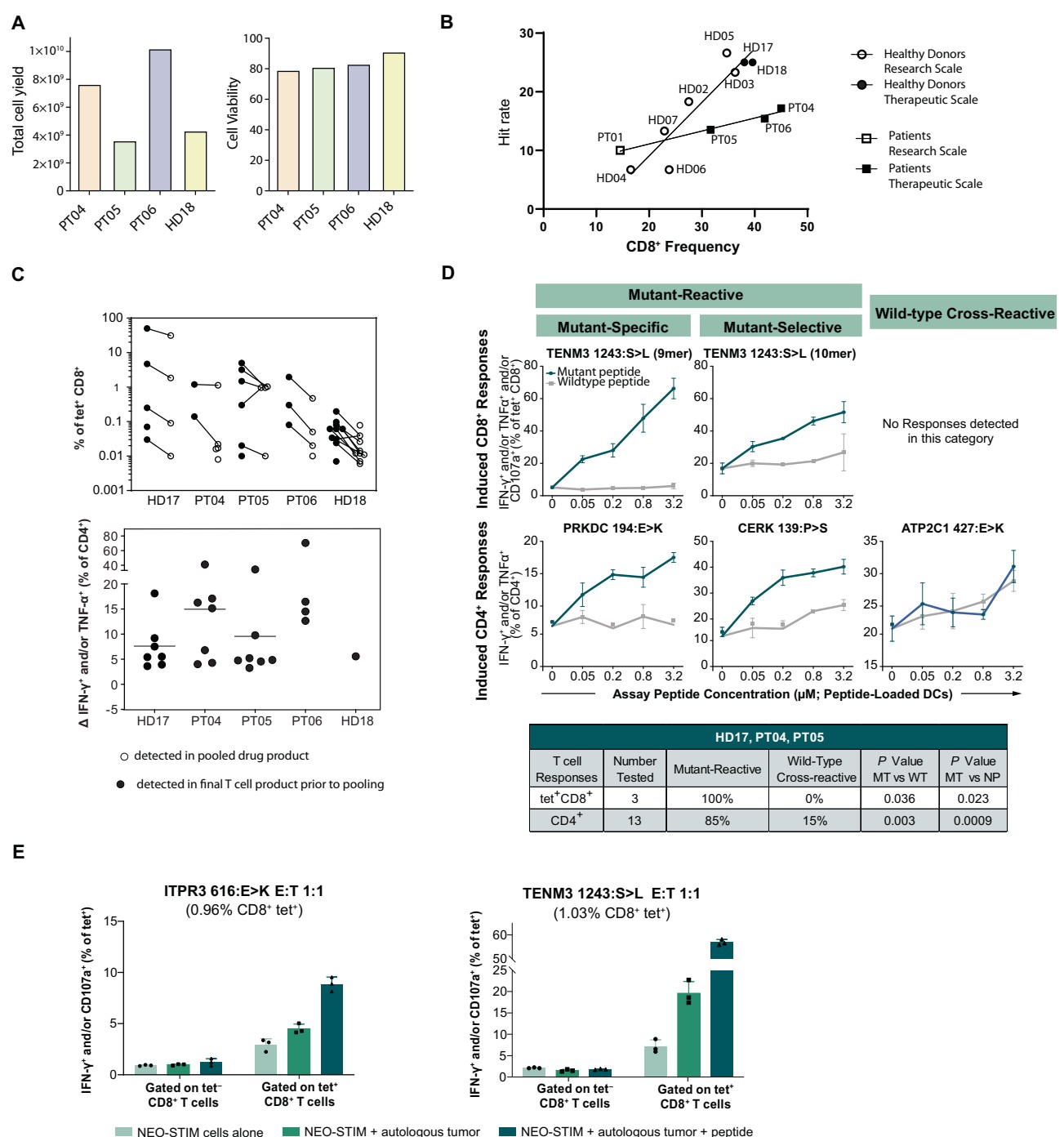

derived single-cell suspension cells and CD8[+] T cells from the DP were co-cultured, and cytokine production and degranulation was measured. We observed that the ITPR3$_{616:E>K}$-specific cells were functional in response to autologous tumor only when additional exogenous mutant peptide was added.

Importantly, the TENM3$_{1243:S>L}$ response showed an even stronger increase over the negative control, which was statistically significant. This recognition was observed without addition of exogenous peptide and thus means TENM3$_{1243:S<L}$-specific cells were able to recognize and respond to physiologically relevant levels of antigen on autologous tumor (Fig. 4E).

Overall, we were able to evaluate autologous tumor recognition by three distinct T cell responses derived from two patients (Figs. 2I and 4E). Two out of the three responses were able to recognize autologous tumor (TENM3 1243$_{S>L}$ and LRBA 2198$_{S>L}$). Interestingly, while a high

variant allele frequency (VAF) and high variant RNA expression did not correspond to observed immunogenicity in our system (Supplementary Data S1), they are likely to be relevant with respect to tumor recognition. Indeed, we observed that the VAF and RNA expression of the mutant ITPR3 616$_{E>K}$ variant (corresponding to the T cell response that was unable to recognize autologous tumor without exogenous peptide loading) was the lowest of the three (Supplementary Data S1). While sample availability was limited in this study, testing the correlation of VAF and RNA expression with tumor recognition in additional samples would be valuable to further confirm these results.

**Neoantigen-specific T cells with distinct phenotypes are recovered from blood and tumor post-infusion of DP**
Early clinical proof of concept has been established with T cell products manufactured through NEO-STIM, and translational data from

**Fig. 4 | Successful manufacturing at therapeutic scale using the NEO-STIM process. A** Total cell yield and viability of the drug products (DPs) formulated on day 26, and using comparable methods. Note that sample from donor HD17 was processed with three stimulations (see Supplementary Data S1), therefore this data point did not fulfill pre-established criteria for analysis, data was not included here. **B** Correlation between frequency of CD8[+] T cells (as a % of live cells post-depletion) and hit rate. Hit rate was defined as the number of actual hits over number of possible hits among six replicates. PT04 was processed at research scale and at therapeutic scale, but the datapoint for research scale was excluded post-analysis, since the method used at research scale was not comparable to the rest of the small-scale runs. $R^2 = 0.87$ for healthy donors, $n = 8$ and $R^2 = 0.98$ for patient samples, $n = 4$. Correlation was performed using simple linear regression method. **C** Frequencies of CD8[+] and CD4[+] responses across $n = 5$ patients. See Supplemental Data S1 Therapeutic Scale for number of responses tested. Each data point represents the frequency of antigen-specific T cells in the final drug product (DP, open symbols) or in the individual vessels of the post-NEO-STIM T cell product prior to pooling (closed symbols). Top: percent tetramer[+] of CD8[+]. See Supplementary Fig. 5 for flow cytometry plots of responses from DPs. Bottom: Delta of IFN-γ[+] and/or TNFα[+] as a % of CD4[+]. CD4[+] analysis was only performed at the vessel level prior to pooling. Horizontal lines represent mean. **D** Example graphs for mutant (MT)-specific, MT-selective and wild-type (WT) cross-reactive responses post-NEO-STIM (as defined in a recall response assay) from HD17, PT04 and PT05 after challenge with DCs loaded with MT or WT neoantigen peptides at different concentrations. IFN-γ and/or TNFα and/or CD107a was measured for tetramer-positive (tet[+]) CD8[+] responses and IFN-γ and/or TNFα for CD4[+] responses. Data is shown as mean with SD ($n = 3$ technical replicates). See Supplementary Fig. 7 for all specificity profiles and text for definition of specificity terms; those from therapeutic-scale DPs are listed in the summary table in this panel at the bottom. *P* values in summary table have been calculated across all donors and responses in the respective T cell population (tet[+]CD8[+] or CD4[+]) for MT vs WT at 0.8 μm (MT and WT) and for MT vs no peptide (NP) at 0.8 μm (MT) vs 0 μm (MT) using a paired two-sided t test. tet[+]CD8[+]: $n = 3$ responses from one donor; CD4[+]: $n = 13$ responses across 3 donors. **E** T cell response against autologous tumor cells. Neoantigen-specific T cells were tested against a single-cell suspension obtained from autologous tumor through a recall response assay (with or without exogenously loaded peptide). Readout: IFN-γ[+] and/or CD107a[+] as % of tetramer[+] CD8[+] and tetramer- CD8 + T cells (*Y* axis). Data shown as mean with SD ($n = 3$ technical replicates). Source data are provided as a Source Data file.

this trial showed that neoantigen-specific responses can be tracked at the antigen level in blood and tumor pre- and post-infusion[45]. Additional translational analysis was performed here for one patient, NAC04[45], evaluating persistence post-infusion down to the level of individual TCR clones. We found that among the pre-existing responses initially detected at the antigen level, some were in fact de novo at the clonotype level, suggesting that a subset of clonotypes may have been induced de novo through NEO-STIM[45]. Across two responses which initially were considered pre-existing, three out of eight subclones were de novo. One of those three was detected post-infusion, compared to five out of five which were pre-existing (Fig. 5A, B). A similar observation of clonal heterogeneity regarding pre-existing and de novo responses was made in another patient from the same trial, NAC01 (Supplementary Fig. 11).

T cells from the peripheral blood of patient NAC04 were single-cell sorted using barcoded tetramer staining and, alongside a post-infusion tumor biopsy digest sample, subjected to single-cell RNA & VDJ sequencing to characterize neoantigen-specific T cell responses. Neoantigen-specificity was defined as peripheral or tumor VDJ sequences matching the validated clonotype from the DP. T cells carrying these responses (NeoAg) were further phenotyped and compared to other cells (Other) from the same sample. We observed an increase in activated/exhausted phenotypes (e.g. *CD86, TNFRSF9, PDCD1, LAG3, HAVCR2*) and cytotoxic phenotypes (e.g. *GZMB, IFNG, PRF1, NKG7*) in neoantigen-specific T cells (NeoAg) versus other cells (Other) in both blood and tumor. In the tumor we also observed increased *CXCL13* mRNA expression on neoantigen-specific T cells, which has been described as a marker for neoantigen-reactive cells and shown to promote tumor infiltration[46–48] (Fig. 5C, D).

The activation status of neoantigen-specific cells versus other cells could only be determined from the blood sample, since the tumor sample did not yield a sufficient number of neoantigen-specific cells to perform this analysis. We observed a significant increase in the activation status of neoantigen-specific T cells compared to other cells in the blood (Fig. 5E).

Taken together, de novo and pre-existing T cell responses were detected post-infusion in both tumor and peripheral blood of a patient, and these responses displayed both activated/exhausted and cytotoxic phenotypes.

## Discussion

Our goal was to develop an advanced personalized ACT approach for cancer immunotherapy, building on recent successes with TIL therapy and addressing some of its main challenges. Key differentiators of NEO-STIM compared to TIL are the systematic generation of a broad set of T cell responses ex vivo, both de novo and pre-existing, using peripheral blood as starting material.

Our data support the critical feature that NEO-STIM can systematically broaden the number of epitopes which can be targeted against the tumor through raising de novo T cell responses, which we conclude based on experiments with material from 23 healthy donors and nine patients (six with melanoma and three with ovarian cancer, representing indications with a mid-to-high mutational load[34]). While "de novo response" is a practical definition for the purpose of the study, a subset of those responses likely stem from the naïve compartment, as healthy donors were able to raise responses against neoantigens they had not encountered before. While in the melanoma patients presented here we only detected one CD8[+] response and zero CD4[+] responses prior to NEO-STIM (likely explained by the assay sensitivity), in another study more than half of CD8[+] responses identified in DPs of patients with melanoma were defined as de novo, using a highly sensitive detection method (bulk TCR sequencing)[45]. In our current work we demonstrate that this number could even be higher, when analysis is done at the clonal level. In addition to increasing the breadth of the anti-tumor response, the de novo induced responses tend to also have a higher activation status compared to pre-existing responses, as shown in this work and in eight additional patients[45]. Together this is expected to result in a more potent T cell response overall.

NEO-STIM manufacturing is initiated with peripheral blood, a starting material that is readily accessible from all patients. Additionally, our data also indirectly supports the previously made observation that T cells from the peripheral blood in general have phenotypic and functional advantages over T cells derived from the tumor[15]. We observed a shift from memory to effector phenotypes and an increase in the Fuchs activation score upon rechallenge with cognate peptide. This indicates that the T cells in the NEO-STIM DPs were not terminally exhausted, although we did not compare DPs made from peripheral blood and tumor directly.

NEO-STIM at both research and therapeutic scale showed robust manufacturability of DPs, suitable for clinical application. Every DP comprised multiple CD8[+] and CD4[+] T cell responses against tumor neoantigens, and in two cases a CD4[+] and CD8[+] response was observed against the same antigen. The latter might be particularly desirable, since CD4[+] cells provide critical helper functions, and intratumoral triads of CD4[+] T cell, CD8[+] T cell, and APC have been associated with anti-tumor response[18,19]. The choice of nDCs as APCs over moDCs might have facilitated this result, since only nDCs were able to raise responses against long peptides. Detailed characterization showed that neoantigen-specific T cells in the DP were polyfunctional and

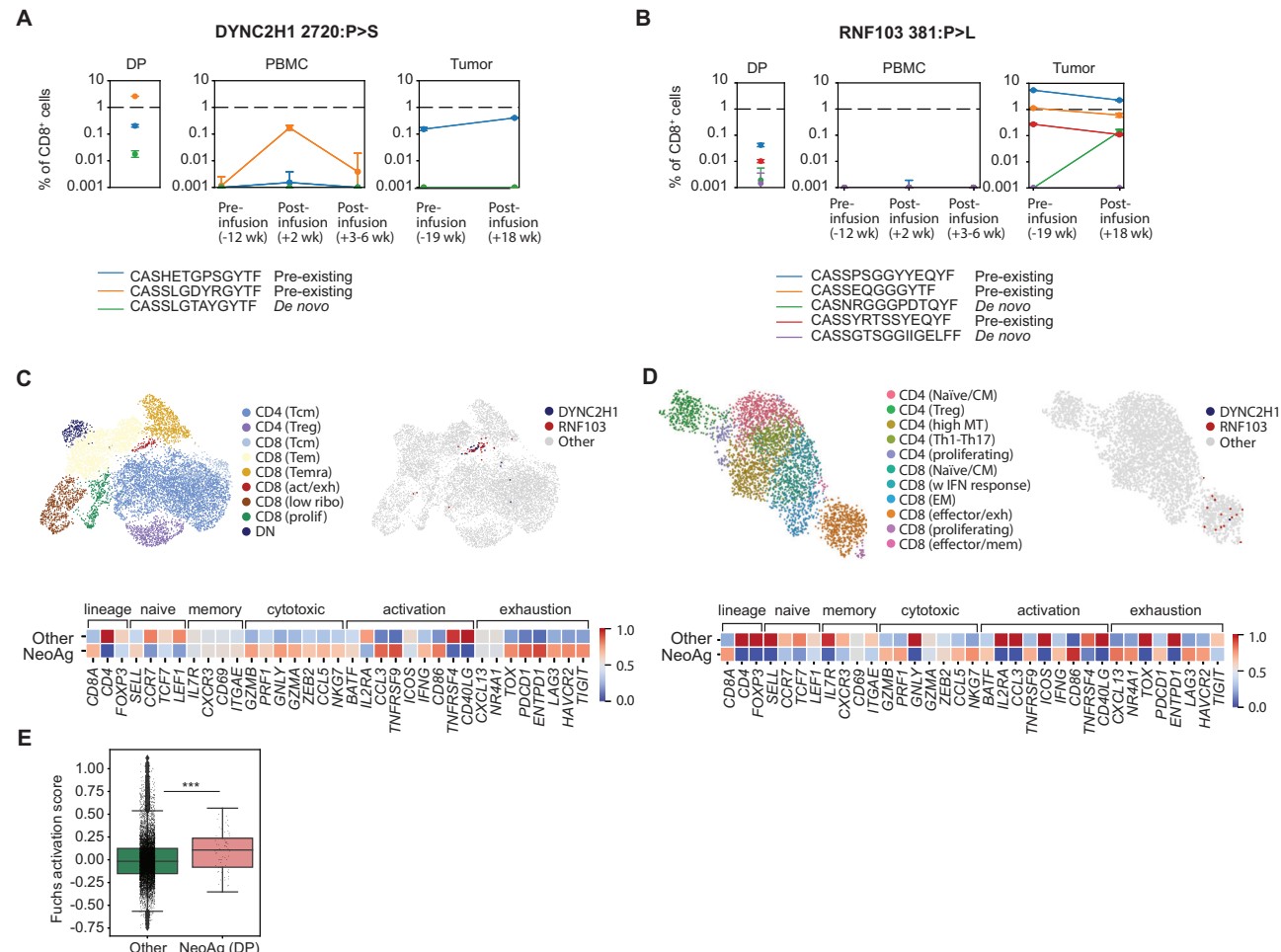

**Fig. 5 | Phenotypic analysis of neoantigen-specific responses post-infusion.**
**A**, **B** Tracking of two different neoantigen-specific responses in the DP, blood and tumor of patient NAC04[45] by bulk TCR sequencing at the clonotype level. These responses were initially sorted on antigen-specific cells (tet⁺ CD8⁺) in DP and sequenced to understand clonotypes at a single-cell level. Data points show mean frequency of clonotypes and standard deviation of $n = 3-5$ different clonotypes (each measured as technical duplicates). **C**, **D** Uniform manifold approximation and projections (UMAPs) of multimodal single-cell RNA/TCR sequencing data from the blood (**C**) and tumor biopsy (**D**) of the same patient. Neoantigen-specific T cells (NeoAg) were labeled if peripheral or tumor TCR sequences matched the validated

clonotypes from the DP. The neoantigen-specific cells primarily map to the CD8⁺ activated/exhausted cluster in peripheral blood and the CD8⁺ effector/exhausted cluster area in tumor. Gene expression analysis between neoantigen-specific (NeoAg) and remainder (Other) cells in periphery or tumor comparing the NeoAg vs. Other cells for selected genes as shown in the heatmaps. **E** Fuchs activation score for NeoAg vs Other cells from the peripheral blood. Boxplots show data quartiles. Outliers were defined using 1.5 times the interquartile range. NeoAg (DP): neoantigen-specific cells that were detected in DP. Statistical analysis was by two-sided Mann-Whitney U test. $n = 8778$ (other cells) and $n = 63$ (NeoAg (DP) cells). ***$P = 0.0009$. Source data are provided as a Source Data file.

specific for MT antigen. Finally, the manufactured drug products (DPs) demonstrated cytotoxic potential against tumor cells engineered to express the patients' specific HLA and neoantigens. While these targets are engineered cell lines, we further supported the relevance of our findings by showing recognition of autologous tumor material for two patients. Expanding that analysis further with additional autologous tumor samples would be valuable to further confirm these results.

The FDA has shared draft guidance for cell therapy manufacturers to support the production of consistent drug products[49], as such the observed heterogeneity of the DPs is a limitation. Although the fraction of CD3⁺ T cells was consistently high, the magnitude of antigen-specific responses varied drastically between donors and across the neoantigens. Dividing up the manufactured peptides into six pools, which are then added to six parallel culture vessels may have helped to prevent immunodominance and generate responses even against low immunogenic peptides, albeit sometimes at low levels. However, the implementation of multiple parallel culture vessels did increase complexity of the manufacturing process.

There are several ways to improve the quantity and quality of the generated T cell responses. Increase of neoantigen-specific T cells in DPs, which has been correlated with better clinical outcome in TIL products[9], could be achieved with antigen-specific T cell enrichment post T cell induction, either through activation markers (e.g. CD137[50,51] or by using pMHC streptamers[52]. Alternatively, depletion of further cell types in the starting material could be envisioned, since we observed a correlation of CD8⁺ cell frequency in starting material with hit rate. Finally, overexpression or deletion of genes that impact beneficial T cell phenotypes and that correlate with in vivo persistence, expansion and improved effector functions could be implemented.

Importantly, translational analysis from a patient treated with a NEO-STIM-based product demonstrated that neoantigen-specific T cell responses seen in the DP were detectable in the blood and tumor post-infusion. An increased frequency of some of these responses was observed, both in blood and tumor, at least temporarily. We were able to separate neoantigen-specific T cells from the patient's blood and tumor, and phenotype them in comparison to cells that were not

specific to neoantigens used in NEO-STIM. This analysis revealed a clear upregulation of markers for cytotoxic response, as well as for activation and exhaustion of the neoantigen-specific T cells.

In conclusion, we have provided proof of concept for NEO-STIM to generate an ACT containing polyclonal and functional CD8[+] and CD4[+] T cell responses against multiple personalized neoantigens. We demonstrate the feasibility of manufacturing this product from patients' peripheral blood at a therapeutic scale. A first-in-human clinical study to assess clinical safety and efficacy of DPs generated with NEO-STIM in patients with advanced and metastatic melanoma (NCT04625205) showed that BNT221, the investigational neoantigen-specific T cell product generated through NEO-STIM, was well tolerated and showed early signals of clinical efficacy[45]. Of the nine treated patients, six had stable disease (SD) as their best overall response, including up to 36 weeks for one patient. Four of these patients with SD showed tumor regression (up to −20% according to Response Evaluation Criteria in Solid Tumors (RECIST) version 1.1) after infusion. The drug product characteristics observed in the FIH study are consistent with the findings described here. Furthermore, evaluation of combination strategies with for example checkpoint inhibitors and/or cytokines is warranted to enhance the potency, expansion and persistence of neoantigen-specific cells further. Thus, the NEO-STIM process described here warrants further development to determine the ability of this ACT product to induce neoantigen-specific responses and a therapeutic benefit in patients with advanced cancers.

## Methods

### Ethics statement
The research described in this study complies with all relevant ethical regulations, see section Patient and healthy donor material for detailed board approval information.

### Study Design
With the goal of developing a robust process to induce de novo and expand existing T cell responses from autologous patient material, the NEO-STIM method was established as follows: Initial setup and optimization was carried out with PBMCs from 20 HDs, using model antigens as immunogens (Supplementary Table 1). Optimized parameters and final protocol are shown in Fig. 1B. The resulting process was then applied to PBMCs from four patients with melanoma and three patients with ovarian cancer (Supplementary Data S1, Supplementary Table 2) at research scale, using patient-specific neoantigens as immunogens. A scale-up of this personalized drug product generation process was achieved with PBMCs from two HDs and three patients with melanoma (Supplementary Data S1, Supplementary Table 2). The experiments at research scale were carried out in biological triplicates (melanoma) or biological quadruplicates (ovarian cancer) for up to seven pools of peptides and at therapeutic scale in single biological replicates for up to six vessels (one vessel per peptide pool). Output parameters to test the performance of the final protocol were number and quality of induced T cell responses and phenotype of resulting T cells. Post-infusion analysis was performed on material from two patients from a clinical trial using NEO-STIM to generate T cell DPs[45].

### Patient and healthy donor material
Leukapheresis was obtained from six individual patients with metastatic melanoma, three patients with ovarian carcinoma and 23 healthy individuals. Among the six patients for process development, three patients (PT01-PT03) were enrolled under the trial NT001 (NCT02897765) conducted by Neon Therapeutics, USA. The study was approved by the Institutional Review Board at each participating site. Six patients (PT04-PT09) participated in the N16NEON trial (CCMO ID: NL57848.031.16) under an institutional review board (IRB)-approved protocol conducted at Netherlands Cancer Institute (NKI). The IRB was the Medisch-Ethische Toetsingscommissie AVL (METC AVL). The two

patients for post-infusion analysis were treated at NKI in a phase I study (NCT04625205) approved by the Central Committee on Research Involving Human Subjects (approval number NL72301.000.19) and DP, PBMC or tumor single cell digest aliquots were provided for this study. All studies were conducted in accordance with the Declaration of Helsinki. All patients signed an informed consent form. Patients did not receive reimbursement for the study participation. HD material was sourced from AllCells, USA or HemaCare, USA.

### Processing of leukapheresis and tumor material
Leukaphereses collected from patients and HDs were processed under universal precautions. PBMCs from NT001 participants and HDs were prepared by Ficoll-paque (GE Healthcare) density-gradient centrifugation and cryopreserved in either Recovery Cell Culture Freezing Medium (Invitrogen)[20] or Human serum (Sigma) with 10% DMSO. Material collected from N16NEON patients was washed using wash media (4% w/v human serum albumin [HSA] and 0.4% w/v sodium citrate in NaCl) and cell pellets were resuspended in 4% w/v sodium citrate in HSA that was frozen by adding an equal amount of freezing media (20% v/v DMSO, 4% w/v sodium citrate in NaCl). Tumor material collected from N16NEON patients was digested into single-cell suspension and frozen using freezing media into cryovials. All processed material was shipped and stored at <−140°C until use. Post-infusion tumor resection or biopsy was collected, processed and stored as described previously[45].

### Identification of patient-specific neoantigens, peptide synthesis
A detailed method of identifying patient-specific neoantigens was described previously[20]. For each patient a set of induction peptides (used for NEO-STIM) and a set of assay peptides (used for read-outs) were synthesized using solid phase peptide synthesis and purified by reverse phase HPLC. Preparative HPLC fractions that met purity and identity criteria based on UPLC-UV/MS analysis were pooled and dried using parallel centrifugal evaporators or lyophilizers. These peptides were resuspended in DMSO and pooled together for experimental use. Synthesis and purification took place at Creosalus, at PPL or at BioNTech.

Induction peptides were either short peptides (8-12 amino acids [aa]) predicted to bind to MHCI or long peptides (25 aa) predicted to bind to MHCII. Induction peptides were either patient-specific neoantigens (up to 40 short and up to 20 long peptides) or model antigens (Supplementary Table 1). Assay peptides (20 aa) were synthesized as overlapping (at least 9 aa overlap) to cover the sequence of the long mutated induction peptide.

### Rapid induction protocol
A rapid induction protocol was used to evaluate the impact of various APC populations on the priming of T cells in NEO-STIM. PBMCs were thawed and resuspended in AIM-V media (Gibco, 12055-083), incubated with Benzonase at 250 U/mL (Sigma-Aldrich 70746) for 30 minutes at 37 °C, and then depleted to remove monocytes (CD14[+]) and T regulatory cells (CD25[+]) according to the manufacturer's protocol (Miltenyi Biotec, Inc 130-050-201, 130-092-983). For research experiments where additional populations were depleted, CD11c-biotin (clone Bu15; Biolegend #337232), CD11b-biotin (clone ICRF44; Miltenyi #301304), and Anti-Biotin microbeads (Miltenyi #130-090-485) were used according to the manufacturer's instructions. Post-depletion cells were cultured in AIM-V media with FLT3L (CellGenix 1415-050) at 50 ng/mL plated in low volume (2 mL) in a 24-well G-Rex plate (Wilson Wolf, 80192 M) for 18-24 h in six biological replicates. Previously identified neoantigens (Supplementary Table 1) in the form of peptides (0.4 μM final concentration) were incubated for 1 hour and the cultures were matured using maturation cocktail with TNF-α (1000 U/mL; CellGenix 1406-050), IL-1β (10 ng/mL; CellGenix 1411-050), Prostaglandin-E 1 (PGE-1; 0.5 μg/mL; Cayman Chemical, 745-65-

3), and IL-7 (0.5 ng/mL; CellGenix 1410-050). The cells were left in culture for 14 days in total with the addition of human serum (HS) 18-24 h post maturation, the addition of media on day 5 with IL-7 (CellGenix 1410-050) and IL-15 (CellGenix 1413-050), and a media replacement on day 12. Exception: For comparison with the moDC protocol, medium was changed on day 5 to AIM-V/RPMI (18% AIM-V + 71% RPMI1640 glutamax + 10% Human serum + 1% PenStrep, all v/v), with IL7 and IL15 additions every two to three days.

## NEO-STIM process using nDCs

Preparation of PBMCs, stimulation and expansion until day 14 were carried out as in the rapid induction protocol with two modifications: Plating was either as 2 mL in a 24-well plate in biological triplicates (research scale) or as 9 mL in a 6-well plate or 10 M or 100 M G-Rex vessel (therapeutic scale; Wilson Wolf), and peptide incubation was 2 h. For the therapeutic-scale experiments, cells were grown in one large culture vessel per peptide pool (10 M or 100M G-Rex vessel), up to six distinct pools per patient. The 2° culture (as well as 3° culture) was generated by thawing unstimulated PBMCs and preparing them as done for the 1° culture. For the process with one restimulation (two stimulations total), the 2° culture was initiated on day 12 (unless otherwise specified) and added to 1° culture on day 14 for a total culture time of 26 days. For the process with three stimulations, the 2° culture cells were initiated on day 12 and combined with 1° culture cells on day 14; 3° culture cells were initiated on day 19 and combined with 1°/2° cultures on day 21, with a 28-day total culture time. (Exception: When comparing nDC-based and moDC based protocols, first restimulation was on day 15, second restimulation on day 22 and harvest on day 29 to be able to compare the two protocols side-by-side). For the large-scale semi-closed system process, the depletion was carried out using a CliniMACS system, and the cells were cultured in G-Rex 10 M or 100 M vessels. The success of the manufacturing process was evaluated using the following acceptance criteria: Cell count $10^9$-$10^{10}$ cells; Viability ≥70% as set forth by FDA guidance[53]; T cell purity/identity (CD3$^+$ cells) ≥40%; Mycoplasma negative; Endotoxin ≤1.25 EU/mL; Sterility: no growth.

## moDC preparation and use in moDC-based NEO-STIM

Monocyte-derived dendritic cells were prepared from PBMCs to stimulate T cells and expand antigen-specific cells. Monocytes were isolated from PBMCs using CD14 microbeads according to the manufacturer's protocol (Miltenyi Biotec, Inc 130-050-201). Isolated monocytes were resuspended in DC medium (CellGenix 20801-0500) and plated with GM-CSF (800 U/mL; CellGenix 1412-050) and IL4 (400 U/mL; CellGenix 1403-050). After 5 days, immature DCs were harvested and incubated with neoantigens in the form of peptides (0.4 μM final concentration) for 1 hour and the cultures were matured using maturation cocktail with TNF-α (10 ng/mL; CellGenix 1406-050), IL-1β (10 ng/mL; CellGenix 1411-050), IL6 (10 ng/mL; CellGenix 1404-050) and Prostaglandin-E 1 (PGE-1; 0.5 μg/mL; Cayman Chemical, 745-65-3) for 2 days. Post maturation differentiated DCs were then cultured in AIM-V/RPMI medium (18% AIM-V + 71% RPMI1640 glutamax + 10% Human serum + 1% PenStrep, all v/v). APCs were used in moDC based NEO-STIM at a ratio of 1:10 (DC to T cells) to stimulate CD14/CD25-depleted PBMCs on days 0, 13 and 20 of the process.

## Dendritic cell analysis

For measurement of costimulatory markers on DCs, DCs were matured as described above and after 2 days were harvested and stained with surface antibodies for 30 min at 4 °C. Cells were washed and resuspended in FACS buffer at 4 °C. To characterize the dendritic cells, they were gated on: Cells, Singlets, Live all, CD3$^-$/CD19$^-$, then mDCs were gated against CD11c$^+$ and HLA-DR$^+$; cDC1: HLA-DR$^+$, CD11c$^+$, CD141$^+$; and cDC2: HLA-DR$^+$, CD11c$^+$, CD1c$^+$. The fraction of CD40$^+$, CD80$^+$, CD83$^+$ and CD86$^+$ expressing cells was quantified, as well as the MFI of those markers using FlowJo software (v07-10).

Corresponding flow panel: Dendritic cell analysis panel (Supplementary Data S2)

## pMHC multimer analysis using tetramers

A combinatorial coding approach using pMHC tetramers to identify neoantigen-specific CD8 T cells was applied, as described previously[54]. Briefly, cells were thawed, washed, and resuspended in phosphate-buffered saline and then stained with the tetramers associated with respective pMHC molecules, in addition to a number of surface markers. This allowed for quantification of the fraction of tetramer$^+$ CD8$^+$ T cells.

In the pMHC tetramer analysis, cells of interest were gated as previously described[45] i) Lineage: Forward scatter and side scatter gates were applied to filter for singlets of the target cell population, PBMC. Dead cells were excluded with a live/dead marker. Cells were then gated for positive CD8 staining and negative CD4/CD14/CD16 and CD19 staining. ii) Single positives: CD8$^+$ cells stained with the single fluorophores, which were used for the tetramer labeling, were gated. iii) Combicoding: Peptide – MHC complexes were labeled with two different fluorophores to make tetramers. T cells that interacted with a defined combination of two and only two of the fluorochromes assigned to a specific peptide-MHC complex and satisfied additional specificity criteria[54] were considered specific to the respective neoantigen.

Corresponding flow panel: Class I tetramer panel (Supplementary Data S2)

## Investigation of cell expansion using proliferation dyes

The proliferation of the cells of interest was investigated using the CellTrace CFSE Cell Proliferation Kit (Thermofisher Scientific #C34554) and the Tag-IT violet proliferation and cell tracking dye (Biolegend, #425101) at final concentrations of 0.5 μM and 2.5 μM respectively. For labeling, the cells were washed and incubated with proliferation dyes for 5 minutes at 37°C and later quenched with five times the volume of 20/80 media (18% AIM-V + 71% RPMI + 10% HS + 1% Penstrep) at room temperature in the dark for 5 minutes. The cells were then washed again and cultured in the NEO-STIM process to track multiple generations of antigen-specific cells, using dye dilution, by flow cytometry. 1° culture cells were harvested on day 14 and labeled with Cell trace CFSE before adding to 2° cultures labeled with Tag-IT violet for restimulation. The antigen-specific cells were identified using combinatorial coding technology (as described in the previous section) and gated on FITC (CFSE) or BV421 (Tag-IT violet) to understand the proliferative capacity of specific populations.

Corresponding flow panel: Proliferation panel (Supplementary Data S2)

## Phenotypic flow cytometry characterization

The subpopulations in starting material, in-process-samples and post-NEO-STIM samples were analyzed using flow cytometry. All samples were subjected to the following gating strategy: live/dead marker$^-$, CD14$^-$/CD16$^-$/CD19$^-$, CD8$^+$ (either bulk or tetramer$^+$ subset) or CD4$^+$. Subsequent analysis of phenotypic markers (CD62L and CD45RA) using FlowJo software (v07-10) was performed to delineate subpopulations based on differentiation status of the T cells as mentioned below.

CD62L$^+$ CD45RA$^+$: Naïve T cells, CD62L$^+$ CD45RA$^-$: Central memory T cells, CD62L$^-$ CD45RA$^-$: Effector memory T cells, CD62L$^-$ CD45RA$^+$: Effector T cells.

Corresponding flow panel: Differentiation status panel (Supplementary Data S2)

## Recall response assay

A multi-color flow cytometric panel in combination with an overnight antigen recall assay in which the NEO-STIM-induced cultures were co-cultured with peptide-loaded APCs was used at a ratio of 1:10 (DC to

T cells) to identify neoantigen-specific CD4[+] T cell responses. Here, monocyte-derived dendritic cells were used as APCs which were differentiated over seven days from CD14[+] monocytes using GMCSF (Stock 800U/μL) and IL-4 (Stock 400U/μL). During the differentiation process, immature DCs were loaded with respective neoantigens in the form of assay peptides at a final concentration of 0.8 μM and matured with TNFα (10 ng/mL), IL1β (10 ng/mL), IL-6 (10 ng/mL), and PGE-1 (0.5 mg/mL)[45]. Pre-NEO-STIM and induced T cells (in process or final NEO-STIM or DP samples) were co-cultured overnight with or without the antigens of interest in the format of overlapping 20 aa assay peptides at a final concentration of 0.8 μM each. After the overnight incubation, a neoantigen-specific CD4[+] T cell response was defined by the increased expression of IFNγ and/or TNFα in the presence of target antigen compared to the negative control (co-cultured with unloaded DCs). In the flow cytometric readout, cells of interest were gated as follows: live/dead marker[−], CD14[−]/CD16[−]/CD19[−], CD4[+]. Subsequently, production of IFNγ or TNFα or both (IFNγ and TNFα) by the CD4[+] T cells was assessed for identifying antigen-specific responses and the difference in frequency of response over background was plotted. For the polyfunctionality profiles, the production of IFNγ, TNFα and CD107a in response to assay peptide-loaded APCs was assessed. FlowJo software (v07-10) was used.

To understand the polyfunctionality profile of the antigen-specific CD8[+] T cell responses, cells were stained with pMHC tetramer molecules and then assessed for the production of IFNγ, TNFα and CD107a in response to assay peptide-loaded APCs. To describe antigen specificity, the responses were categorized based on differential reactivity towards MT or WT neoantigen peptides, measured at different concentrations (0, 0.05, 0.2, 0.8 & 3.2 μM). Significance of differences between peptide-stimulated cells and no peptide control, as well as between MT and WT reactivity was calculated using paired t-test with FDR correction for adjusted $P$ value or Tukey multiple comparison test or Sidak's multiple comparison test, as indicated in the Source Data file. Exact $P$ values were recorded for all comparisons in the Source Data file.

Corresponding flow panel: Recall response panel (Supplementary Data S2)

## Cytotoxicity assay

Neoantigen-expressing tumor lines were generated by transducing A375 tumor cells using pCDH_CMV_MART1-T2A-eGFP vector with the relevant HLA (https://www.ebi.ac.uk/ipd/), as well as with neoantigen- (or corresponding WT-) epitope, and flanking sequence. By transducing a sequence that includes the flanking region, the epitope can only be presented if the antigen presentation machinery of the cell is capable of processing the sequence. Separately, for a subset of the specificities tested, A375 tumor cells with the HLA of interest were loaded with the relevant peptide. The induced T cells were then co-cultured with these tumor lines. After 6 hours, the tumor and T cells were analyzed by flow cytometry using FlowJo software (v07-10). The upregulation of active caspase 3 (an early apoptotic marker on tumor cells), and mobilization of CD107a (expressed when T cells degranulate) were measured. The apoptotic target cells were gated on live/dead marker[−], Cas3[+] and gating for CD8[+] T cells: live/dead marker[−], CD3[+], CD8[+] CD107a[+].

Corresponding flow panel: Cytotoxicity assay panel (Supplementary Data S2)

## Tumor recognition assay

A single-cell suspension of autologous tumor corresponding to the patients PT04 & PT05 and T cells generated through NEO-STIM from each of these patient samples were co-cultured overnight. In the flow cytometric readout, cells of interest were gated as follows: live/dead marker[−], CD14[−]/CD16[−]/CD19[−], CD8[+] and pMHC[+]. Subsequently, functional markers CD107a and IFNγ were quantified among

antigen-specific (pMHC[+]) CD8[+] T cells using FlowJo software (v07-10).

Corresponding flow panel: Recall response panel (Supplementary Data S2)

## Functional Avidity Assay

Functional avidity of neoantigen-specific TCR clones was determined using an NFAT reporter Jurkat model as described previously[45]. Briefly, E6.1 Jurkat T cells (ATTC; TIB-152) engineered with an NFAT luciferase reporter were used to detect T cell activation by electroporating them with the respective TCRα and TCRβ chain RNA. TCR-expressing Jurkat cells and peptide-loaded A375 cells (ATTC; CRL-1619) were co-cultured at 1:1 effector-to-target ratio. After overnight co-culture luminescence was determined and NFAT activation per condition was calculated by normalizing luminescence to a negative control reaction containing no peptide. Estimated $EC_{50}$ is reported based on fit curve analysis for cases, where two peptide concentrations showed significant NFAT activation.

## Single-cell RNA sequencing and TCR seq analysis

**Preparation of neoantigen-specific T cells from NEO-STIM, post-infusion blood and tumor samples for single-cell RNA/TCR sequencing.** Neoantigen-specific CD8[+] T cells from NEO-STIM samples were activated through a recall response assay and were sorted on BD Aria using MHC class I tetramers. Single-cell RNA and TCR libraries were prepared using the Single-cell 5′ VDJ Reagent kit (10x Genomics). The library preparation process was performed according to the associated 10x Genomics protocol, including GEM Generation & Barcoding, Post GEM-RT Cleanup, cDNA Amplification, Target Enrichment from Amplified cDNA, 5′ Gene Expression Library Construction, Enriched Library Construction, and Sequencing Libraries. The sequencing was performed on a NovaSeq 6000 system (Illumina).

Post-infusion patient PBMCs were thawed and subjected to T cell enrichment by using a human Pan T Cell Isolation Kit (Miltenyi Biotec #130-096-535). Isolated T cells were counted and treated with Benzonase and 50 nM Dasatinib for 20 minutes at 37 °C in RPMI media containing 10% FBS. Enriched T cells were centrifuged at 476g for 5 minutes, washed once with FACS buffer (1X PBS + 0.5% w/v BSA) supplemented with 50 nM dasatinib, and plated in a 96-well plate (3×10⁶ cells/well). Cells were stained with the barcoded tetramers associated with a pooled tetramer mix, followed by surface markers staining (BV711 anti-human CD3 (BD) antibody), and LIVE/DEAD Fixable Near-IR Dead Cell Stain Kit (Thermo Fisher #L34975), allowing for quantification of the fraction of neoantigen-specific T cells. After 30-minutes of incubation at 4 °C, cells were washed twice in PBS with 1% BSA and sorted using an MA900 cell sorter (Sony Biotechnology). For a given patient sample, two different T cell populations were sorted: Tetramer[−] (Live CD3[+]) and tetramer[+] (Live CD3[+] dual tetramer[+]). After sorting, labeled cells were resuspended at -1×10³ cells/ml in PBS supplemented with 2% FBS and submitted to subsequent single-cell RNA sequencing (scRNA-Seq) processes. Sample processing for single-cell gene expression (GEX) and TCR V(D)J clonotypes (VDJ) libraries was performed following the manufacturer's protocol (Chromium GEM-X Single-Cell 5′ Reagent Kits v3, 10X Genomics). All libraries were incorporated with Sample Index sequences (Dual Index Kit TT Set A) and sequenced on the NovaSeq 6000 system (Illumina). The sequencing parameters were: Read 1 of 28 cycles, i7 Index of 10 cycles, i5 Index of 10 cycles, and Read 2 of 90 cycles.

Tumor biopsy sample was kept on ice and enzymatically and physically dissociated using a Human Tumor Dissociation Kit (Miltenyi, 130-095-929) following manufacturer's protocol. Briefly, tumor biopsy fragments were minced into small pieces using scalpels and mixed with an enzyme mix (Enzyme H, R, and A) into a gentleMACS C Tube (Miltenyi). The C tube was then placed onto the sleeve of the GentleMACS Dissociator and an appropriate GentleMACS Program was chosen according to tumor type. After dissociation, the

sample solution was filtered through a 70 μm cell strainer to collect the dissociated cells with complete RPMI 1640 medium passing through the 70 μm cell strainer. Filtered tumor cells were then rested in complete RPMI medium containing 10% FBS and 1% penicillin/streptomycin for 3 hours at 37 °C. After resting, cells were washed two times, counted, resuspended in PBS, supplemented with 2% FBS, and submitted to subsequent single-cell RNA sequencing processes. Sample processing for single-cell GEX and VDJ libraries was performed following the manufacturer's protocol as described above.

Corresponding flow panel: Sequencing sort panel-NEO-STIM, Sequencing sort panel-Post-infusion, CITE panel (Supplementary Data S2)

**Computational analysis of single-cell RNA/TCR sequencing data.** Data was aligned using Cell Ranger (10x Genomics) with the following versions and reference assemblies. NEO-STIM samples were aligned with Cell Ranger 6.0.1, GEX reference GRCh38-2020-A, and VDJ reference GRCh38-alts-ensembl-7.0.0, using the 'multi' option. Post-infusion blood samples were aligned with Cell Ranger 8.0.1, GEX reference GRCh38-2024-A, VDJ reference GRCh38-alts-ensembl-7.1.0, and a custom CITE panel of antibodies using the 'multi' option. Tumor samples were aligned using Cell Ranger 7.1.0 (GEX reference GRCh38-2020-A, 'count' option) and Cell Ranger 7.0.1 (VDJ reference GRCh38-alts-ensembl-7.1.0, 'vdj' option). Python package Scanpy 1.9.1[55] was used for downstream analysis of data acquired from Cell Ranger.

Filtering was performed as follows: We removed cells that had (1) more than 10% mitochondrial gene content, (2) fewer than 1000 UMI (NEO-STIM, tumor) or 2000 UMI (post-infusion) from RNA, and (3) fewer than 300 genes (NEO-STIM, tumor) or 1500 genes (post-infusion) from RNA. For blood samples, cells were also removed if they had (4) fewer than 800 CITE UMI or (5) fewer than 20 CITE markers detected. For blood and NEO-STIM samples, we also filtered to include only protein-coding genes, as annotated in Gencode Basic 4.2, that were expressed in at least 0.1% of all cells.

RNA libraries were log2-normalized following library-size normalization with a scale factor of $10^4$; with the exception of tumor digests, which were $\ln(X + 1)$ normalized using sc.pp.log1p. These values were used for visualization of gene expression, including heatmaps and UMAP plots, and identification of highly variable genes. The 4000 most variable genes were detected by the sc.pp.highly_variable_genes function (flavor=Seurat), with the tumor analysis utilizing a 'min_mean=0.02', 'max_mean=4', and a 'min_dispersion=0.5'. For blood and NEO-STIM samples, these 4000 genes were overlaid with 10x Genomics Immunological panel (1056 genes) to de-noise the list.

Downstream processing was performed on each patient individually to limit batch and donor effects. For integration across samples within each patient, data was normalized as follows. RNA libraries were natural-log-normalized following library-size normalization with a scale factor of $10^4$, then scaled with the sc.pp.scale function using a max value of 10. Total counts were regressed out using the sc.pp.regress_out function. CITE libraries were CLR normalized and then scaled with the sc.pp.scale function using a maximum value of 10. Principal component analysis (PCA) was performed using the sc.tl.pca function. For blood and NEO-STIM samples, we performed multiomics factor analysis (MOFA) using the muon library function mu.tl.mofa with 10 factors to integrate the RNA and protein data after restricting to highly variable genes as defined above[56,57]. Nearest neighbors were found using the sc.pp.neighbors function: for blood and NEO-STIM, we utilized the 10 factors from MOFA and for tumor digest we utilized the top 25 principal components. Then, UMAP dimensionality reduction was performed with sc.tl.umap to obtain a two-dimensional representation of the cell states. For clustering, we used the Scanpy sc.tl.leiden function.

**TCR library preparation and bulk TCR-seq.** Library preparation and sequencing was performed as described previously[45]. In brief, TCRβ libraries were prepared from isolated RNA derived from the snap-frozen T cell pellets (1 million pan T cells each) or tumor. Amp-RepSeq was used for PBMC, DP and tumor samples. Libraries were sequenced using a NextSeq Reagent Kit v2 300 cycles (Illumina) according to the manufacturer's protocol.

**TCR repertoire generation.** TCR repertoires were generated by applying MiXCR 4.3.2 software on the paired-end raw sequencing FASTQ files. TCRβ CDR3 clonotypes were filtered by removal of non-functional sequences (out-of-frame sequences, those containing stop codons or those of length ≤5 aa). Clonal frequency was calculated based on the read count for each clone out of the total read count.

**Calculation of T cell activation scores.** We used a published T cell activation signature by Fuchs et al.[44], found in Table S5 (tab 'TOP50-separators expression') in that publication representing the top 50 genes upregulated in tetramer-sorted antigen-specific CD8+ T cells after stimulation with cognate antigen (influenza or CMV). Each cell was assigned an activation score representing the average Z-score for the gene set using log2(library-size normalized) expression values.

### Statistical Analysis
For continuous variables, comparisons of means between groups were made using t-tests or ANOVA, unless stated otherwise in Figure captions or respective tabs in the Source Data File. Correlation analysis between two quantitative variables was performed using simple linear regression and measured based on $R^2$ value to determine the goodness of fit. All analyses were performed using GraphPad versions 7.04-9.5.1 or Python version 3.9.

### Reporting summary
Further information on research design is available in the Nature Portfolio Reporting Summary linked to this article.

## Data availability
DNA and RNA sequencing data are not publicly available for contractual obligations. Based on ethical approval: the data can be made available upon request for future research purposes involving adoptive cell therapies for the treatment of advanced cancers; it however cannot be used for publications outside of this study. All data provided are anonymized to respect the privacy of patients who have participated in line with applicable laws and regulations. Data requests pertaining to the manuscript may be made to the corresponding author (M.M.v.B., marit.vanbuuren@biontech.us). Requests will be processed within 16 weeks. The remaining data are available within the Article, Supplementary Information or Source Data file. Source data are provided with this paper.

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

## Acknowledgements

We thank the patients for donating PBMCs and tumor material for continued development of adoptive cell therapies. We thank the BNT research and development teams, both current and former employees of BioNTech US and Neon Therapeutics: Yvonne Ware, Yuting Huang, Daniel Kallin, Zhengping Huang and Khristiana Jones, Julia Papps, Diana Velez, Prerna Suri, Rachel DeBarge, Jonathan McGee, Zakaria Khonder, Dewi Harjanto, Paul Turcott, Janani Sridar, Ed Fritsch, Luke Henry and Matt Goldstein. We thank Myriam Armant for support and advice on upscaling activities (employee of Boston Children's Hospital). The authors received editorial assistance from Acumen Medical Communications. The study was sponsored and funded by BioNTech.

## Author contributions

Program strategy: M.M.v.B., J.B.H, M.DM., J.H.v.d.B., T.N.M.S. and R.B.G. Conceptualization: D.L., J.K., B.MC., M.N., N.A.M.B., R.D.B., J.A., F.D.B., J.H.F.S., K.B., M.DM., A.P., M.R., J.Z.D., J.R.S., V.J., C.M.A, C.M.N., B.N., K.M., T.N.M.S., R.B.G., J.B.H., J.H.v.d.B., M.M.v.B. Methodology: D.L., J.K., B.MC., M.N., N.A.M.B., R.D.B., E.K.J., J.A., F.D.B., J.H.F.S., S. Hannes, K.S., Z.X., K.B., E.E., S. Hymson, S.M., M.V.Z., S.S., B.R., A.P., C.M.N., J.H.v.d.B., M.M.v.B. Software: E.K.J., E.E., K.S., Z.X., A.P. Validation: D.L., J.K., B.MC., N.A.M.B., R.D.B., E.K.J., J.A., J.H.F.S., K.S., C.G., K.B., E.E., M.V.Z., S.S., B.R., A.P., M.R., J.Z.D., J.R.S., V.J., C.M.A, C.M.N. Formal Analysis: D.L., J.K., B.MC., M.N., N.A.M.B., R.D.B., E.K.J., J.A., F.D.B., J.H.F.S., S. Hannes, K.S., Z.X., K.B., E.E., S. Hymson, S.M., A.P., C.M.N., J.H.v.d.B., M.M.v.B. Investigation: D.L., J.K., B.MC., M.N., N.A.M.B., R.D.B., J.A., F.D.B., J.H.F.S., K.S., C.G., K.B., M.V.Z., S.S., R.V., B.R., C.M.N. Resources: D.L., J.K., B.MC., N.A.M.B., R.D.B., J.A., E.K.J., K.S., M.V.Z., S.S., B.R., J.S.W.B., M.W.R, C.M.N., J.B.H., J.H.v.d.B. Data Curation: D.L., J.K., E.K.J., J.A., F.D.B., J.H.F.S., S. Hannes, K.S., C.G., K.B., E.E., A.P., J.H.v.d.B., M.M.v.B. Writing – Original Draft Preparation: D.L., J.K., E.K.J., E.E., C.G., M.M.v.B. Writing – Review & Editing: all authors Visualization: D.L., J.K., B.MC., N.A.M.B., R.D.B., E.K.J., F.D.B., J.H.F.S., C.G., K.B., E.E., A.P., J.H.v.d.B., M.M.v.B. Supervision: D.L., J.K., R.B.G., J.B.H., J.H.v.d.B., M.M.v.B. Project Administration: D.L., J.K., N.A.M.B., R.D.B., C.G., C.M.N., J.B.H., J.H.v.d.B., M.M.v.B.

## Competing interests

J.S.W.B., and M.W.R. declare no competing interests. E.K.J., S.Hannes., C.G., S.M., K.N.B., A.P., M.R., J.Z.D., J.R.S., V.R.J., C.M.A., R.B.G., and M.M.v.B. are employees of BioNTech US or BioNTech SE. D.L., J.K., B.MC., M.N., E.E., K.S., J.H.F.S., J.A., Z.X., F.D.B., M.DM., K.M., and S. Hymson are former employees of BioNTech US or Neon Therapeutics. T.N.M.S. is a founder of Neon Therapeutics. D.L., J.K., B.MC., M.N., E.K.J., C.G., J.A., E.E., S.M., K.S., K.N.B. A.P., M.R., J.Z.D., J.R.S., V.R.J., C.M.A., M.DM., K.M., T.N.M.S., R.B.G., M.M.v.B. own stocks of BioNTech. N.A.M.B., R.D.B., M.V.Z., S.S., R.V., B.R., C.M.N., B.N. were under contract with BioNTech for drug product manufacturing. D.L., J.K., N.A.M.B., R.D.B., V.R.J., C.M.A., M.DM, T.N.M.S., J.H.v.d.B. and M.M.v.B. are inventors of patent applications that cover part of this article. NEO-STIM patents are pending in various countries (WO2020227546A1 and WO2019094642A1, - "T cell manufacturing compositions and methods"). J.B.H. received grant or research support from BioNTech US, Bristol Myers Squibb, Novartis AG, Amgen, Sastra Cell Therapy, Asher Bio. J.B.H. discloses ownership interest in Neogene Therapeutics. J.B.H. has an advisory role for Agenus, Astra Zeneca, Bristol Myers Squibb, CureVac, GSK, Imcyse, Iovance Bio, Immunocore, Ipsen, Merck Serono, MSD, Molecular Partners, Novartis, Orgenesis, Pfizer, Roche/Genentech, Sanofi, Third Rock Ventures and is a Scientific Advisory Board (SAB) member at Achilles Tx, BioNTech US, Instil Bio, T-Knife, Neogene Therapeutics (AZ) and Sastra Cell Therapy. R.B.G is on the Board of Directors for Alkermes and Zai Lab and is a member of the SAB for Leap Therapeutics Inc. T.N.M.S. declares advisory roles for Allogene Therapeutics, Asher Bio, Merus, Neogene Therapeutics and Scenic Biotech, is stockholder in Allogene Therapeutics, Asher Therapeutics, Cell Control, Merus and Scenic Biotech, and is venture partner at Third Rock Ventures. BioNTech's role in the study is disclosed in the author contributions, none of the other organizations had a role in the study or publication.
