## [Transparent Peer Review file · Nature Communications]

NEO-STIM advances personalized neoantigen-specific adoptive T cell therapy

Corresponding Author: Dr Marit van Buuren

Version 0:

Reviewer comments:

Reviewer #1

(Remarks to the Author)

This manuscript provides a detailed description of optimization of a process for generating tumor antigen reactive T cells by in vitro sensitization of peripheral blood cells for use in clinical adoptive cell transfer trials. In addition, characterization of the cells administered to a single melanoma patient and the corresponding cells that appeared to persist following ACT provides data that was informative. There were several points, however, that were not adequately addressed in the manuscript.

1. While infused TIL products may in some cases only appear to contain a small percentage of tumor reactive T cells, the frequencies of those cells are much higher than the 0.5% in the cited references, for example in Parkhurst et al. *Cancer Discov.* 9:1022-1035, 2019.

2. The logic of combining a culture receiving a 10 peptide stimulation with a 20 culture or even a 30 stimulation was not clear. It appears that a portion of the 10 culture was pulsed with the appropriate peptide in a 20 stimulation and then the pulsed cells were mixed at varying ratios with the remaining cells from the 10 culture. Was this done in an attempt to minimize the activation induced cell death that may have occurred in the re-stimulated cultures or was there another reason for employing this somewhat complex protocol? Why weren't unstimulated autologous PBMC pulsed with peptide and used for the 20 and 30 stimulations? Also does the hit rate correspond to the frequency of reactive T cells or the % of antigen reactivities that were generated?

3. While the protocol that was developed appears to have utilized an initial 14 day stimulation period followed by 1 or 2 stimulations for 7 days each, were studies carried out to determine the optimal timing of the subsequent stimulations?

4. Responses against MELANA are highly atypical in that they can readily be generated in the majority of A*02:01+ patients and healthy donors and so should not be used to model neoantigen responses.

5. The criteria used to categorize antigens as highly or weakly immunogenic should be provided. In addition, an explanation of the reasoning behind dividing antigens into these 2 categories should be provided.

6. The experiments evaluating the effectiveness of DCs in stimulating T cells were carried out by depleting CD11b and CD11c populations but not by enriching these populations. If CD11c cells are the most effective at priming T responses then positively selected cells should be tested, as these would then presumably be the most effective at generating T cell responses. In addition, the data demonstrating that CD11c+ cells were needed to generate class I responses when stimulating with long peptides should have been shown.

7. The tetramer staining results for some of the specificities shown in Fig. S5 need additional explanations. For example, the single tetramer populations for DMN2 S88:P>V, NLRP1 123:E>K, ZNF226 777:H>Y and PANCM 2031:D>H appeared to be comparable to or higher than the double positive populations. Given the fact that a substantial proportion of the cells were positive with a single fluorophore, how confident can you be that the double positive populations were antigen specific?

8. The sentence beginning on line 154 states that 'Responses that were not detected prior to NEO-STIM were called de novo responses, although we cannot rule out that at least some of those responses were present in the starting material at low frequencies, below the detection limit of the assay.' This is not accurate, as T cells must have been present at some frequency prior to stimulation. The T cells may have been expanded from the naïve or memory T cell pool, which may have been what was meant by this statement, but this can be determined by separating naïve and memory T cells prior to deep TCR sequencing, which can be carried out in 106 or more PBMC. Another question is whether T cells have been expanded in a patient relative to a healthy donor bearing the same HLA restriction element, but given the differences in the T cell repertoire present in multiple individuals, this could probably only be carried out using tetramer analysis, which would be difficult due to the extremely low frequencies of the majority of reactive T cells.

9. The evaluation of CD4 reactivity by evaluating intracellular cytokine responses may have under-represented the frequency of reactive T cells. Additional assays such as measurement of CD4 reactivity should have been carried out by evaluating upregulation of 4-1BB, OX-40 or CD154.
10. The results of the analysis of specificity for some of the mutant epitopes were unclear. In Fig. 2G and 2H there was essentially no difference between the response of ARAP 828:Y>H T cells in response to cells pulsed with the mutant vs. the wild type peptide or tumor cells transduced with the mutant or wild type construct and the frequencies of cells reactive with the mutant peptide and construct were quite low. The specificity of the CSNK1A1 27:S>L reactive T cells was also not clearly demonstrated by the results shown in Fig. 2H where only a minimal difference was observed between responses to tumor cells transduced with the mutant and wild type constructs. In addition, it is not correct to claim that the response of ITPR3 616:E>K responses are tumor specific, as there was no significant increase in the IFN-g response when cells were cultured with tumor cells.
11. It is not clear what was meant by the statement beginning on line 266 that 'This further highlights the importance of cell subset depletion at the beginning of the process.' Were CD4+ and CD8+ T cells separated before that peptide stimulation was carried out?
12. The criteria used to define neoantigens as being mutant specific, mutant selective or wild-type cross-reactive are unclear. The reactivity against the CD8 neoepitope TENM3 1243:S>L appears to be mutant specific, as the baseline or reactivity was ~20% and the response to the wild type peptide was only slightly elevated at the highest peptide concentration. Responses such as the CD4 response against the ATP2C1 427: E>K CD4 epitope should not be considered when evaluating these responses as they are unlikely to recognize any naturally processed and presented antigen.
13. An explanation is needed for the disparities between the relatively high frequencies of some of the CD8 reactivities shown in Fig. 2 and Fig. S5 and the data shown in Fig. 4 where all the frequencies of all of the reactivities, with the exception of 1, presumably that directed against the SRSF1 33:E>K neoepitope, are below 2%. The data presented in Fig. 4C also needs further explanation. Why are all of the points for CD4 reactivities filled but only 1 of the CD8 specificities filled? Do the unfilled points represent the final concentration of these reactivities in the pooled product, and why are some reactivities shown as being in the T cell product before pooling but not in the final product, as all of them should have been present unless they were not pooled? An explanation is also needed for the data in the supplementary table marked 'Therapeutic scale' (should this be labeled Supplementary Table 2?). What is meant by 'vessel', or 'well', and why are the frequencies sometimes higher in the vessel than in the DP whereas sometimes the reverse is true? How was the final DP generated – were all of the wells containing neoantigen reactive T cells combined, and were there significant differences between the expansions carried out using the different peptide pools? Since multiple wells containing the same peptide pools were set up, were all of the wells pooled, or only the wells that showed effective expansion of reactive T cells pooled, and if so what criteria were used?
14. In general, did de novo/pre-existing reactivity correspond to specificity categories of mut specific/selective, wt-reactive? For DYNC2H1 or RNF103 reactivities, were multiple TCR clonotypes observed in the scRNA data (Fig 5C/5D)? If so, were there distinctions in phenotypic state distribution within or among clusters and did these differ by de novo or pre-existing categories?

Reviewer #2

(Remarks to the Author)

This manuscript describes the development of NEO-STIM, a platform that induces de novo and expands memory T cell responses to patient-specific neoantigens ex vivo from peripheral blood. The authors convincingly demonstrate that the resulting drug products (DPs) are polyclonal, polyfunctional, mutant-selective, and cytotoxic, with clear recognition of autologous tumor. Their platform overcomes several limitations of TIL-based ACT, including reliance on resectable tumors and exhausted phenotypes, and is supported by preclinical and early clinical feasibility data. All major conclusions are well supported, and the methodological detail is commendable. However, several revisions would improve clarity and reproducibility:

Major Concerns:

1. Please explicitly state which subsets are depleted (e.g., CD14⁺ monocytes, CD25⁺ Tregs), whether CD11c⁺ DCs are enriched, and what maturation cocktail is used. Consider including a side-by-side comparison of DC phenotypes and costimulatory marker expression after maturation to validate the superiority of naturally circulating DCs over moDCs.
2. The magnitude of neoantigen-specific responses varies widely across patients. If feasible, correlate DP response magnitude with pre-culture CD8⁺ T cell frequency or baseline TCR repertoire richness. At minimum, please discuss whether predictive biomarkers (e.g., mutation burden, CD8⁺/CD4⁺ ratio, memory subset composition) might inform response yield and product quality.
3. The authors report a minority of responses with WT reactivity. Please summarize the percentage of tested responses falling into each specificity category (mutant-specific, selective, WT-crossreactive). If possible, add data on functional avidity (e.g., EC50 curves) for at least one WT-crossreactive versus mutant-specific clone to support safe discrimination. Clarify whether WT-reactive clones would be excluded from clinical release criteria.
4. PRDM1 induction after repeated stimulation suggests terminal effector skewing. The authors might consider comparing CD62L⁺CCR7⁺ memory phenotypes across 1x vs 2x stimulation DPs to assess whether reduced stimulation could preserve less-differentiated states. A brief analysis of exhaustion marker expression (e.g., PD-1, LAG-3) in the final product would also be informative.

5. In the patient with tracked TCR clonotypes, the activated/exhausted phenotype is well described. If possible, include a quantitative comparison of persistence or expansion between pre-existing and de novo clonotypes. This could clarify the relative durability of newly induced versus expanded memory clones.

6. Would this approach work in tumors with fewer mutations (e.g. some epithelial cancers or pediatric cancers)? It likely can, as long as at least a handful of neoantigens are identifiable, but the effort (60 peptides per patient) might yield fewer hits. A sentence discussing this (perhaps noting that even low-mutational-load tumors can still have a few strong neoantigens to target) would be useful to frame the breadth of this therapy.

7. Additionally, while the focus is lab/process development, the authors did treat at least one patient and mention an ongoing Phase I trial. If possible, the clinical context should be expanded slightly: For example, the authors state BNT221 showed early signals of efficacy – without divulging too much beyond what is allowed, can they say what signals? (Tumor regression in some patients? Durable T cell persistence?) Even a general statement like “early signs of anti-tumor activity were observed in the trial, such as objective tumor responses in some participants” would underline the importance. If such details are not yet public, it’s understandable, but then perhaps emphasize that a formal clinical trial is underway to assess safety/efficacy, which is a strength of this work’s translational readiness.

8. I recommend adding a sentence on how this approach might integrate with other therapies – for instance, will patients receive checkpoint inhibitors alongside NEO-STIM T cells? This could be relevant since ex vivo expanded cells might still benefit from checkpoint blockade post-transfer. Even if not part of this study, acknowledging combination strategies (as is common with TIL therapy + IL-2 or checkpoints) would show awareness of the clinical landscape.

Minor Concerns:

9. It’s excellent that each peptide pool was run in triplicate to test process reproducibility. The supplemental data likely show high correlation (the text notes $R^2=0.98$ between some replicates). The authors might consider mentioning in the Results that reproducibility between parallel cultures was high, as this will give readers confidence that the process yields consistent results. A brief mention of the replicate consistency (perhaps referencing Supplementary Fig. S5 or S1B as appropriate) would suffice.

10. In the main figures, some data (e.g., Fig 2A examples, 2H cytotoxicity, 2I tumor recognition) seem to highlight one or two patients. It would be helpful if the authors could ensure that data from all four patients are reported, at least in aggregate. For example, Fig. 2C, 2E summarize the number of responses per patient – presumably each dot is one patient’s result. Clarify in the figure caption if those are per-patient medians or individual responses. Similarly, for cytotoxicity and tumor recognition, if not all patients were tested, state the number of patients or responses evaluated. Transparency about n for each assay will help readers gauge the consistency across the cohort.

11. Overall, the figures are well-designed. A few suggestions: (i) Figure 1B – if this is a schematic of the process, ensure that all steps are labeled clearly (e.g., “Deplete PBMC subsets,” “Primary stimulation (Stim1) day 0–14,” etc.). Currently it’s referenced that Fig. 1B shows process steps; just make sure it’s readable and perhaps include an explanation in the legend for any icons or abbreviations. (ii) Figure 2 – panels C, E (showing range and medians of responses) should indicate whether the percentages shown are fraction of total CD8 or CD4 T cells that are neoantigen-specific (I assume yes). Adding “% of CD8⁺ T cells” in the axis label or caption would help. Also, define “tet+” in the caption for non-specialist readers (T cells detected by peptide-MHC tetramers). (iii) Figure 3 – the UMAP plots and heatmaps from scRNA-seq are complex. The authors might add a brief description in the legend about what the major clusters represent (activated effector vs memory, etc., as derived from Fig. S7). A simplified summary of the key takeaway (that ARAP1 de novo clone shifts to effector state early, vs the SRSF1 memory clone which shifts gradually) could guide the reader through panel E/F.

12. Please ensure all abbreviations are defined on first use. For instance, “DP” (drug product) is defined in the Introduction – that’s good. Also define Stim1/2/3 clearly as rounds of stimulation. The term “research scale” vs “therapeutic scale” is used; perhaps clarify that “research scale” refers to small-scale lab process used for initial 4 patients, and “therapeutic scale” is a full-scale manufacture (if I understood correctly). Minor point: “WT” and “MT” (wild-type vs mutant) should be defined in figure legends or methods for clarity.

13. The authors cite relevant literature on TIL therapy and neoantigen vaccines. One suggestion is to cite recent clinical successes: for instance, the FDA approval of Lifileucel for melanoma in 2024, which they allude to in the introduction, could be explicitly referenced as a milestone that underscores the importance of advancing neoantigen-targeted T cell therapies. Additionally, if there are any comparable approaches (perhaps a recent study attempting to prime T cells from blood for adoptive therapy, if any), mentioning those would position this work in the landscape. It appears this approach is quite unique, but for completeness, the authors might consider if groups have tried similar “prime-and-expand” T cell therapies (I recall some early trials expanding T cells to tumor-associated antigens or virus antigens ex vivo). A sentence distinguishing NEO-STIM from TCR-engineered cell therapies would also help: e.g., emphasize this does not require genetic modification, making it potentially faster and less expensive once neoantigen peptides are identified.

Reviewer #3

(Remarks to the Author)

This study develops and validates NEO-STIM, an ex vivo method to generate personalized neoantigen-specific T-cell

responses under optimized conditions that avoid immunosuppression.

Lenkala et al. validated the protocol using model antigens with healthy donor PBMCs expressing relevant HLA types. They enhanced T cell priming by depleting inhibitory cell subsets and identified the best APC populations by assessing expansion, diversity, and functionality of CD8+ responses—including those to long peptides that usually induce CD4+ T cells. Applying the method to PBMCs from four melanoma patients, they detected both pre-existing and newly induced CD8+ responses, via combinatorial pMHC tetramer staining – as well as CD4+ T cell responses. The generated DP T cells were further phenotypically and functionally characterized. Finally, the protocol was optimized for therapeutic-scale manufacturing while maintaining functional activity.

The study addresses an important and unmet clinical need with a promising approach. The idea is compelling and has strong potential for personalized immunotherapy. However, the work feels preliminary and would benefit from more extensive analysis and deeper characterization to strengthen its conclusions and clinical relevance.

Major concerns:

- Limited donor and patient diversity: Proof-of-concept experiments use only two healthy donors (one with weak de novo reactivity), with very low response frequencies (<1%), raising questions about functional relevance. The clinical cohort is restricted to melanoma patients, nearly all previously treated with immunotherapy, limiting generalizability and potentially confounding immune responses.
- Lack of tumor antigen expression and clonality data: The manuscript does not report expression levels or clonal distribution of the selected neoantigens on patient tumors, which is critical for evaluating clinical impact.
- Insufficient functional validation and controls: Functional assays with autologous tumor cells are limited (only two responses tested), and killing assays are lacking. Healthy autologous cells were not used as specificity controls. Many experiments rely on single replicates, and controls often lack APC plus non-mutated peptide conditions, weakening robustness and specificity assessment.
- Use of artificial tumor models: Validation frequently relies on tumor cells transduced to express the desired neoantigens, which may not reflect physiological expression or clinical relevance. Confirmation with authentic autologous tumor cells is necessary.
- Incomplete phenotypic and CD4+ T cell characterization: Phenotypic analysis is largely restricted to CD8+ T cells and compares only one pre-existing and one de novo response. There is no thorough characterization of other CD8+ or CD4+ T cell responses. Additionally, no pre-protocol assessment of CD4+ responses was conducted to distinguish pre-existing from induced de novo populations, limiting interpretation of efficacy.

Minor concerns:

- Proof-of-concept responses should be shown separately for high and low immunogenic antigens to support claims on low immunogenic antigen recognition.
- Activation markers should be shown separately for clarity.
- Supplementary Figure S1E is hard to interpret and uses different donors than the previous experiments, making it difficult to get an overall picture.

Version 1:

Reviewer comments:

Reviewer #1

(Remarks to the Author)

The majority of issues raised by the reviewers were addressed in the revised manuscript; however, there were a few additional points that need to be clarified.

In response to question 5 from reviewer 1, the distinction between highly immunogenic and weakly immunogenic still seems artificial. Highly immunogenic antigens might simply possess a larger repertoire of reactive clonotypes and therefore might elicit a more robust response. While this is of some interest, the most important point regarding tumor antigens is their relative potency against natural tumor antigen targets, as this can vary widely depending on the target.

In response to question 12 from reviewer 1, responses were defined as 'mutant-selective' if there was significant reactivity against the mutant epitope but 'some' reactivity against the wild type epitope. This is not a clear definition, and some criteria should be used such as a 10-fold or greater difference between the response to the mutant and wild type peptide. It should also be pointed out that wild-type cross-reactive T cells will be ineffective at tumor control as these presumably represent targets that are not naturally processed and presented at levels on tumor cells.

The response to question 7 from reviewer 2 was unclear. The statement that was added to the text indicated that six had stable disease as their best overall response but four of these patients showed tumor regression. Why was the best overall response of these four patients not designated tumor regression? Was tumor regression based upon RECIST criteria, and what was the duration of responses seen in these patients?

Reviewer #2

(Remarks to the Author)

The revision and rebuttal are careful, specific, and, on the whole, responsive. Most of my major and minor concerns are fully or substantially addressed with added methods detail, clearer figure legends, new supplemental analyses (specificity categories; functional avidity), and expanded clinical context and discussion. Only two items remain partially addressed (noted below). Explicit clinical release criteria for WT-cross-reactive clones, and breadth of claims for low-TMB tumors. Otherwise, the manuscript is materially clearer and stronger.

1. The authors now summarize the distribution mutant-specific 7/19 (36.8%), mutant-selective 9/19 (47.4%), WT-cross-reactive 3/19 (15.8%) and provide full profiles in Suppl. Fig. 7, with example panels and a summary table in Fig. 4D. They also added functional avidity data for four TCRs in Suppl. Fig. 8, which was exactly the kind of orthogonal support I suggested. However, my request to clarify whether WT-reactive clones are excluded from clinical release is not answered explicitly; the rebuttal focuses on bulk-product infusion and post-hoc assessments. The new data are strong; a single sentence stating the release criterion will fully close this point.

2. The new ovarian cancer data demonstrate feasibility beyond melanoma and are presented transparently; however, the reported TMB 25–88 spans moderate–high mutational loads, so it doesn't fully speak to the lowest TMB contexts I had in mind. Still, adding these data and updating counts strengthens generalizability.

Reviewer #3

(Remarks to the Author)

The authors have provided a careful and detailed rebuttal, and several of my earlier points have been appropriately addressed. In particular, the revisions related to donor inclusion and diversity (Question 1), antigen immunogenicity classification (Question 6), and figure and marker clarity (Questions 7 and 8) improve the manuscript's transparency and readability. These efforts demonstrate the authors' commitment to strengthening the presentation and completeness of their work.

However, several core scientific concerns remain insufficiently resolved. Regarding Question 2, the inclusion of tumor antigen expression and clonality data represents only a minimal and largely cosmetic fix. The new data are not analyzed or discussed, merely appended to a supplementary spreadsheet. There is no examination of how antigen expression correlates with immune responses, nor any commentary on tumor purity or clonal context. While this addition improves transparency, it adds little biological or clinical insight. For Question 3, the key issue of functional validation remains. The authors continue to rely primarily on engineered A375 tumor cells, which are artificial targets that do not fully reflect native tumor antigen processing or presentation. No new functional data were added, and the rebuttal mainly reinterprets existing results rather than addressing the underlying experimental limitation. Similarly, for Question 4, the justification provided for the use of artificial tumor models is pragmatic but not a solution. Citing another study from the group offers useful context but does not compensate for the lack of validation within this work. The connection is indirect and reads more as an appeal to external support than as additional evidence. For a mechanistic or preclinical study, this remains a significant shortfall. With respect to Question 5, the clarification regarding CD4⁺ T-cell baseline controls is appreciated, but no new data have been added. The CD4⁺ component remains underexplored. While the limitation of available multimer reagents is understandable, reviewers would generally expect alternative or creative approaches to strengthen this aspect in a high-impact submission. Overall, while the revised manuscript is clearer and more complete in presentation, the experimental depth and validation required to fully support the central claims are still lacking.

Version 2:

Reviewer comments:

Reviewer #2

(Remarks to the Author)

I am fully satisfied with the responses to my critiques, and I have no further comments.

Response to Referees

Contents

Reviewer 1 feedback:	2
Reviewer 2 feedback:	17
Reviewer 3 feedback:	27
References:	31

Reviewer 1 feedback:

This manuscript provides a detailed description of optimization of a process for generating tumor antigen reactive T cells by in vitro sensitization of peripheral blood cells for use in clinical adoptive cell transfer trials. In addition, characterization of the cells administered to a single melanoma patient and the corresponding cells that appeared to persist following ACT provides data that was informative. There were several points, however, that we not adequately addressed in the manuscript.

Question 1:

While infused TIL products may in some cases only appear to contain a small percentage of tumor reactive T cells, the frequencies of those cells are much higher than the 0.5% in the cited references, for example in Parkhurst et al. Cancer Discov. 9:1022-1035, 2019.

In the paragraph starting in line 54 (all line references refer to the updated clean versions) we originally stated that one of the key challenges of TIL therapy is “*the relatively low number of pre-existing responses against only 0.5% of tumor neoantigens on average*”. The cited 0.5% frequency refers to the fraction of gene products encoded by somatic nonsynonymous mutations that were naturally immunogenic in the context of melanoma, not the percentage of tumor reactive T cells in TIL. We recognize that this number can be higher in other tumor indications, and we revised the manuscript accordingly, including the Parkhurst reference.

> **Change** (lines 54-55):

“only a relatively low number of tumor neoantigens (0.5-2.3%) being generally recognized by T cells⁹⁻¹⁴.”

Question 2:

The logic of combining a culture receiving a 1^o peptide stimulation with a 2^o culture or even a 3^o stimulation was not clear. It appears that a portion of the 1^o culture was pulsed with the appropriate peptide in a 2^o stimulation and then the pulsed cells were mixed at varying ratios with the remaining cells from the 1^o culture. Was this done in an attempt to minimize the activation induced cell death that may have occurred in the re-stimulated cultures or was there another reason for employing this somewhat complex protocol? Why weren't unstimulated autologous PBMC pulsed with peptide and used for the 2^o and 3^o stimulations? Also does the hit rate correspond to the frequency of reactive T cells or the % of antigen reactivities that were generated?

We confirm that unstimulated autologous PBMCs pulsed with peptide were used to restimulate and expand the primary and/or secondary cultures (see below, updated Figure 1B and updated methods). The restimulation of primary cultures with autologous depleted PBMCs aims to further expand already primed antigen specific T cell responses and prime additional T cell responses with the objective to produce a drug product that is enriched for multiple neoantigen specific T cell response.

The hit rate is used to quantify the proportion of tested epitopes that elicited a response during the NEO-STIM process. For example, a hit rate of 12% means that of the 60 neoantigens included in a clinical manufacturing scale process, T cell responses were detected towards

seven neoantigens in the final drug product. Hit rate does not refer to the frequency of reactive T cells, which is quantified as a percentage of tet⁺ CD8⁺ cells. Hit rate is defined in the respective Figure captions: Suppl. Fig. 3C-G, Suppl Fig 4E-F. Additionally, we added a definition to Suppl. Fig. 1A.

> **Changes:**

We have adjusted the main text (paragraph starting in line 93), the Methods (paragraph starting line 709) and caption of Figure 1B (lines 1003-1004) to clarify the protocol, and we added a definition for hit rate to the caption of Suppl. Fig. 1A (line S-33).

Question 3:

While the protocol that was developed appears to have utilized an initial 14 day stimulation period followed by 1 or 2 stimulations for 7 days each, were studies carried out to determine the optimal timing of the subsequent stimulations?

To clarify, the final protocol described in this manuscript (Figure caption 1B) is a 14-day 1^o culture, which is combined with a two day old 2^o culture and further cultured for 12 additional days, resulting in a total culture time of 26 days. The changes described in the previous question are intended to make this clearer.

As to optimization of time frames, the antigen specific fraction was monitored over time for both the 1^o and 2^o culture (Suppl. Fig. 1D and Suppl. Fig. 2D, E), and the following was observed:

- 1^o culture: While MART-1 (a model antigen we used to mimic the expansion of memory T cell responses, due to its high precursor frequency) reached its highest magnitude 12 days after the start of culture, the T cell responses induced towards high and low immunogenic peptides continued expanding up to the end of the experiment (day 14; Suppl. Fig. 1D, left and right, respectively).

- 2^o culture: The frequency and total number of antigen-specific cells peaked at day 26, i.e. 12 days post restimulation, for MART-1 and also for some of the responses to high and low immunogenic peptides (Suppl. Fig. 2D, E, respectively).

Taken together, these data supported the 14 d (1^o culture) plus 12 d (post restimulation culture) = 26 d (total culture) times. For increased clarity, we changed the axis labels to read “Days after start of culture”.

> **Changes:**

Supplemental Figure 1D: x-axis label, and Supplemental Figure 2D, E: x-axis labels and caption of Suppl. Fig. 2D-E.

Question 4:

Responses against MELANA are highly atypical in that they can readily be generated in the majority of A*02:01+ patients and healthy donors and so should not be used to model neoantigen responses.

We appreciate the reviewer's comment regarding the HLA-A02:01 restricted MELANA/MART-1 responses in HLA-A02:01+ healthy donors. We agree with the assessment that responses to MELANA / MART-1 can be readily generated, since both healthy donors and patients often have MELANA-specific T cells. Since we wanted to look at both pre-existing (memory) and de novo responses, we used MELANA/MART-1 to model memory responses.

Our interest in pre-existing (memory) T cell responses towards neo-antigens is based on the fact that they are sometimes observed in patients¹⁻⁵ and may play an important role in the anti-tumor response^{4,5}. Therefore, we wanted to ensure that our process is equipped to both expand pre-existing T cell responses and induce de novo T cell responses.

The frequency of MELANA⁺ cells has proven to be a valuable benchmark, as it aligns with the frequencies observed during expansion of the memory response SRSF1_{33:E>K} (PT01, 72.1% of CD8⁺ T cells, Fig. 2B, C).

Question 5:

The criteria used to categorize antigens as highly or weakly immunogenic should be provided. In addition, an explanation of the reasoning behind dividing antigens into these 2 categories should be provided.

We appreciate the reviewer's request for clarification regarding the criteria used to categorize antigens as highly or weakly immunogenic, as well as the rationale behind this classification.

Model epitopes were initially selected based on published evidence demonstrating that T cell responses were detected in patients expressing the mutated or viral antigens from which these epitopes are derived⁶⁻⁸. These references show that these antigens are naturally processed and presented in vivo, making them suitable candidates for the development of NEO-STIM.

To refine this selection, we conducted in vitro priming experiments, which revealed distinct differences in the ability of these epitopes to elicit T cell responses. Specifically, epitopes were categorized as 'high immunogenic' if they consistently induced robust T cell responses across the majority of experimental replicates. Conversely, epitopes were classified as 'low immunogenic' if they elicited modest T cell responses sporadically across replicates.

The decision to divide antigens into these two categories was driven by the need to better understand the functional differences between epitopes with varying immunogenic potential. This classification enabled us to systematically analyze the growth kinetics and response dynamics associated with each category, ensuring that experimental decisions were optimized for the unique characteristics of highly and weakly immunogenic epitopes. By stratifying antigens in this manner, we aimed to provide a more nuanced framework for evaluating their immunological behavior and therapeutic potential.

> **Change:** This information has been added in the manuscript (lines 86-89).

Question 6:

The experiments evaluating the effectiveness of DCs in stimulating T cells were carried out by depleting CD11b and CD11c populations but not by enriching these populations. If CD11c cells are the most effective at priming T responses then positively selected cells should be tested, as these would then presumably be the most effective at generating T cells responses. In addition, the data demonstrating that CD11c+ cells were needed to generate class I responses when stimulating with long peptides should have been shown.

We thank the reviewer for their question. The reason we chose depletion experiments over enrichment experiments is that there is overlap in CD11b and CD11c expression. In our work we were considering moDCs and conventional DCs (cDCs). cDCs primarily consist of three main subtypes that express both markers to varying extents as laid out in the Table below⁹⁻¹¹.

With this overlap in expression, enrichment experiments result in the inclusion of the other subset, which is not the case for depletion experiments. While additional human DC markers were used to evaluate the presence of each DC subset, enrichment experiments based on those were out of scope given the low frequencies of them in peripheral blood.

	cDC1	cDC2	pDC
CD11b	Negative > low expression	Low expression	Negative
CD11c	Low expression	Positive	Negative

Despite the abovementioned caveat we still performed the suggested experiment where CD11c+ cells were enriched (n=3 healthy donors). The results are shown here and corroborate our claims in the manuscript.

Description of experiment:

CD11c+ cells were positively selected from the starting material as follows: Freshly thawed PBMCs were depleted of CD14+ and CD25+ cells using Miltenyi microbeads (#130-050-201, 130-092-983 respectively). The CD14-CD25- PBMC were subjected to CD11b+ positive selection (Miltenyi 130-049-601). The resulting CD14-CD25-CD11b- flow through material was used to select CD11c+ cells (Miltenyi 130-113-586, 130-090-485). The CD11c+ fraction which was isolated through positive selection was used to increase or “enrich” the CD11c+ proportion of cells in the CD14-CD25- material, through addition to a new aliquot of CD14/CD25 depleted cells, such that 0.5 million cells from this fraction were added to 5 million CD14-CD25- cells. The NEO-TIM protocol was carried out for base condition (CD14/25 depleted) and for the enriched condition (CD14/25 depleted and CD11c+ enriched).

Results:

Indeed, the hit rate for the high and low immunogenic antigens increased in two out of three donors, when adding CD11c+ cells. However, the MART-1 responses did not benefit from adding additional CD11c+ cells, as quantified by number of antigen specific cells. This may be due to the already high precursor frequency of MART-1-specific T cells in PBMCs of healthy donors,

which could mean neither the precursor T cells nor the DCs are a limiting factor for MART-1 responses.

A, Calculated frequency of CD11⁺ cells in the base condition and after addition of the CD11c⁺-enriched fraction; *B*, Absolute number of MART-1-specific peptides after NEO-STIM for both conditions (the hit rate was 100%); *C*, Hit rate for High and Low immunogenic peptides after NEO-STIM for both conditions.

With respect to the criticality of CD11c⁺ cells for the generation of class I responses with long peptides: these data were included in the original manuscript in the following places:

- Supplementary Figure 3G, (panel F in original manuscript), row 7
- Supplementary Data S1 under the tab Research scale. Note: A filter function in this Excel file under the headers “Responses detected in well” (previously “CD4⁺ responses detected in well”) and “Additional CD8⁺ responses” allows for easy retrieval.

For clarity, the previous header “CD4⁺ responses detected in well” has been changed to “Responses detected in well” , since CD8⁺ responses are also listed under the same header.

>Change: Supplementary Data S1, tab Research scale: Change header “CD4⁺ responses detected in well” to “Responses detected in well”

Question 7:

The tetramer staining results for some of the specificities shown in Fig. S5 need additional explanations. For example, the single tetramer populations for DMN2 S88:P>V, NLRP1 123:E>K, ZNF226 777:H>Y and PANCM 2031:D>H appeared to be comparable to or higher than the double positive populations. Given the fact that a substantial proportion of the cells were positive with a single fluorophore, how confident can you be that the double positive populations were antigen specific?

We appreciate the reviewer’s concern regarding the specificity of the tetramer staining results and the confidence in identifying antigen-specific CD8⁺ T cells. To ensure the robustness and accuracy of our data interpretation, we employed a combinatorial coding strategy, as described by Andersen et al.¹². This approach utilizes a boolean gating method, where each peptide-MHC (pMHC) tetramer is uniquely labeled with a pair of fluorophores, creating a distinct “barcode”

for each epitope. Only T cells staining positively for both fluorophores - and exclusively those fluorophores - are considered antigen-specific.

To further validate the identification of bona fide antigen-specific responses, we applied the following three criteria, as specified by Andersen et al.¹²:

1. The frequency of double-positive cells must reach a threshold of at least 0.005% of CD8⁺ T cells.
2. A minimum of 10 events must fall within the double-positive gate.
3. Visual inspection of the staining ensures that the mean fluorescence intensity (MFI) of both pMHC multimer reagents is approximately equal, resulting in cells consistently aligning along the diagonal in flow cytometry plots.

Single-positive staining derived from T cell responses that share a specific pMHC-fluorophore are also plotted alongside double-positive populations to support visual inspection and confirm staining specificity. This rigorous approach ensures that the tetramer staining exclusively identifies antigen-specific CD8⁺ T cells, providing confidence in the data interpretation.

This is described in Methods, line 784-786. The Figure below from Borgers et al.¹³ further illustrates the gating strategy.

[editorial note: figure redacted]

Supplementary Figure 8 from Borgers et al.¹³:

“Lineage: Forward scatter and side scatter gates are applied to filter for singlets of the target cell population PBMC. Dead cells are excluded with a live/dead marker. Cells are then gated for positive CD8 staining and negative CD4/CD14/CD16 and CD19 staining.

Single positives: CD8⁺ cells stained with the single fluorophores, which are used for the tetramer labelling, are gated.

Combinatorial coding: Peptide – MHC complexes are labelled with two different fluorophores to make tetramers. Double positivity for both fluorophores that have been assigned to a specific peptide in the combinatorial coding process are considered positive for the respective neoantigen. Example:” 0.29% of CD8⁺ cells are specific for peptide 3, and “2.45% of CD8⁺ cells are specific for peptide #7.”

>Change:

Point to additional specificity criteria in Methods line 785-786: *“T cells that interacted with a defined combination of two and only two of the fluorochromes assigned to a specific peptide-MHC complex and satisfied additional specificity criteria⁵⁰ were considered specific to the respective neoantigen.”*

Question 8:

The sentence beginning on line 154 states that ‘Responses that were not detected prior to NEO-STIM were called de novo responses, although we cannot rule out that at least some of those responses were present in the starting material at low frequencies, below the detection limit of the assay.’ This is not accurate, as T cells must have been present at some frequency prior to stimulation. The T cells may have been expanded from the naïve or memory T cell pool, which may have been what was meant by this statement, but this can be determined by separating naïve and memory T cells prior to deep TCR sequencing, which can be carried out in 10⁶ or more PBMC. Another question is whether T cells have been expanded in a patient relative to a healthy donor bearing the same HLA restriction element, but given the differences in the T cell repertoire present in multiple individual, this could probably only be carried out using tetramer analysis, which would be difficult due to the extremely low frequencies of the majority of reactive T cells.

We appreciate the reviewer’s insightful comments to highlight the lack of precision for the definition of de novo responses. We now define this more clearly at first use in the Results (line 114-117):

“De novo responses were defined as responses that were not detected in the pre-NEO-STIM sample. This definition is practical for the purpose of this study as an operational definition, as responses could either originate from T cells in the naïve compartment or from very small memory clones that were below the detection limit of the assay.”

To expand on the role the assay sensitivity plays in this, we also adjusted the discussion. In this study, we attempted to detect CD8⁺ responses pre-NEO-STIM using pMHC tetramers in a combinatorial fashion; additionally, in our clinical study¹³, bulk TCR sequencing was employed, which is also very sensitive with a detection limit of 1:335,000. Based on the observations of both datasets we believe a subset of responses that are not detected prior to NEO-STIM stem

from the naïve compartment. Accordingly, we changed the text in the Discussion (lines 375-381) to:

“While “de novo response” is a practical definition for the purpose of the study, a subset of those responses likely stem from the naïve compartment, as healthy donors were able to raise responses against neoantigens they had not encountered before. While in the melanoma patients presented here we only detected one CD8⁺ response and zero CD4⁺ responses prior to NEO-STIM (likely explained by the assay sensitivity), in another study more than half of CD8⁺ responses were defined as de novo, using a highly sensitive detection method (bulk TCR sequencing)¹³.“

Following the reviewer’s proposal to sort cells into naïve and non-naïve prior to bulk TCR sequencing, we executed such an experiment with cells from one patient, NAC01, from the clinical trial¹³.

However, given that only a subset of T cells can be analysed from a human subject, there is the following caveat: We likely can only substantiate that responses which were labelled as pre-existing indeed stem from the memory compartment. On the other hand, it is unlikely to robustly detect a naïve T cell from a pre-determined response from 10⁶ PBLs due to the scarcity of naïve T cells, which in 95% of cases have a clone size of one^{14,15}.

Accordingly, we learned from the experiment that all five responses tested (all of which had been labelled as pre-existing) appeared to stem from the memory compartment. In addition, we confirmed heterogeneity at the clonal level, i.e. a combination of de novo and pre-existing clones within a neoantigen-response, as we already had described in Figure 5.

Description of the experiment:

Bulk TCR sequencing in naïve and non-naïve T cells was carried out for one patient, NAC01. In this patient, we had detected eight CD8⁺ T cell responses in the drug product (DP)¹³, as shown below. Five of these responses (MRPS30 173:S>F; EP400 3016:A>G; TNPO3 428:K>I; S100A6 62:R>W; LIMA1 401:E>A) were sufficiently robust to facilitate high quality single cell sequencing and TCR reconstruction. All of those had originally been labelled as pre-existing.

[editorial note: figure redacted]

Data from Borgers et al.¹³: Excerpt from Supplementary Figure 2

Three samples from this patient were subjected to bulk TCR sequencing: 1x10⁶ bulk cells from the DP and two 1.28x10⁶ samples of pre-NEO-STIM PBMCs (naïve and non-naïve). To separate naïve from non-naïve T cells, we chose bead-based separation over flow cytometry-based sorting because large cell numbers can be rapidly processed and commercial (validated) kits

are available for separation. We first isolated pan-T cells, then used the CD3⁺ fraction derived from this, to positively select non-naïve cells, leaving naïve cells in the flow through (Miltenyi Biotec kits 130-096-535; 130-097-095). The caveats are that – particularly during the step where non-naïve cells are bead-isolated – cells could potentially be lost. Further, the naïve vs non-naïve separation is based on CD45RO expression, so exhausted T cells and T_{SCM} cells of the respective clone could show up in the naïve-labelled fraction.

The sequencing data demonstrated high quality, identifying approximately 2,500, 25,000, and 25,000 TCR clonotypes in the DP, naïve, and non-naïve PBMCs, respectively, with robust coverage indicated by approximately 9k reads per sample.

Results:

The results were as expected. Five out of five neoantigen-responses were detected in the non-naïve compartment. For response S100A6 all subclones of the response were detected in the non-naïve compartment, validating the methodology we chose. Two of those were also seen in the naïve compartment, likely exhausted cells of the respective subclone. For MRSP30, only two of the eight subclones were detected in the non-naïve compartment, which labels the remaining six as de novo response subclones per our definition, and corroborates results shown in Figure 5 (i.e. some subclones of a response are pre-existing and some de novo).

Bulk TCR sequencing results for drug product (DP), naïve and non-naïve fractions (separated based on CD45RO expression).

The sequencing data was included in the manuscript as Suppl. Fig. 11.

We understand that the second question the reviewer raises relates to whether patients would raise similar responses as healthy donors with the same HLA restriction element. We took a similar approach in this study, by using a patient mutanome to perform epitope predictions in the context of the HLA restriction elements of the healthy donor. Those predicted epitopes were then used in NEO-STIM to raise T cell responses towards neoantigens in healthy donors who were not previously exposed to them.

Generally speaking, both donor types efficiently raised responses, and for both a strong correlation was observed between hit rate and fraction of CD8⁺ T cells in the starting material. However, higher hit rates were observed overall with the HD-derived samples (Fig. 4B).

Additionally, a dataset presented by our group systematically evaluates the immunogenicity of mutant KRAS in the context of common HLA's. These data show that a modified NEO-STIM protocol can prime and expand T cells to KRAS G12 neoantigens effectively from healthy donor material and that TCRs cloned from the process are comparable to clinically efficacious, patient derived benchmarks (Fig. 4B and Fig. 6 in Conn et al.¹⁶).

>Changes:

Add definition of de novo response at first use in Results (lines 114-117); reword discussion for de novo and pre-existing response (lines 375-381); addition of Suppl. Fig. 11

Question 9:

The evaluation of CD4 reactivity by evaluating intracellular cytokine responses may have under-represented the frequency of reactive T cells. Additional assays such as measurement of CD4 reactivity should have been carried out by evaluating upregulation of 4-1BB, OX-40 or CD154.

We appreciate the reviewer's suggestion regarding alternative methods for assessing CD4 reactivity. While we acknowledge that activation-induced marker (AIM) assays may identify a broader fraction of antigen-specific CD4 T cells^{17,18}, our approach prioritized functional cytokine-producing subsets², as these are most likely to provide critical support to CD8 T cells in the tumor microenvironment or lymphoid compartments. Additionally, a third methodology, using pMHC multimers remains challenging to produce for the diverse range of MHCII alleles, limiting their feasibility¹⁹. Our focus on functional readouts aligns with the goal of identifying clinically relevant CD4 T cell responses.

> Change:

The following sentence was added to the manuscript (lines 193-194): "*A functional readout was used with the goal to identify clinically relevant CD4⁺ T cell responses.*"

Question 10:

The results of the analysis of specificity for some of the mutant epitopes were unclear. In Fig. 2G and 2H there was essentially no difference between the response of ARAP 828:Y>H T cells in response to cells pulsed with the mutant vs. the wild type peptide or tumor cells transduced with the mutant or wild type construct and the frequencies of cells reactive with the mutant peptide and construct were quite low. The specificity of the CSNK1A1 27:S>L reactive T cells was also not clearly demonstrated by the results shown in Fig. 2H where only a minimal difference was observed between responses to tumor cells transduced with the mutant and wild type constructs. In addition, it is not correct to claim that the response of ITPR3 616:E>K responses are tumor specific, as there was no significant increase in the IFN-g response when cells were cultured with tumor cells.

We thank the reviewer for their thoughtful comments and have revised the manuscript to improve transparency regarding the specificity analysis.

1. ITPR3 616:E>K Tumor Specificity:

We acknowledge that ITPR3 616:E>K-specific T cells required exogenous mutant peptide to recognize autologous tumor cells, which we have further clarified in the text (lines 328-329).

2. **ARAP1 828:Y>H and CSNK1A1 27:S>L Responses:**

The low response magnitudes in Figures 2G and 2H reflect the use of bulk drug products and gating on bulk CD8⁺ T cells, which resulted in low E:T ratios. Despite this, statistically significant differences between mutant and wild-type responses were observed, underscoring the antigen specificity of these T cells even under suboptimal conditions. The bulk nature is highlighted in the caption of Figure 2G (“...then IFN- γ ⁺ and/or TNF α ⁺ and/or CD107a⁺ secretion of total CD8⁺ T cells was measured).

> **Changes:**

Added clarification (lines 328-329): “*We observed that the ITPR3_{616:E>K} specific cells were functional in response to autologous tumor cells only when additional exogenous mutant peptide was added.*”

Question 11:

It is not clear what was meant by the statement beginning on line 266 that ‘This further highlights the importance of cell subset depletion at the beginning of the process.’ Were CD4⁺ and CD8⁺ T cells separated before that peptide stimulation was carried out?

We appreciate the reviewer’s question and have clarified this statement. The reference to cell subset depletion pertains to the removal of CD14⁺ monocytes and CD25⁺ regulatory T cells prior to peptide stimulation on day 0. This step is critical for enhancing priming efficiency by reducing inhibitory cell populations and increasing the frequency of CD8⁺ T cells in culture. As shown in Fig. 4B, this positively correlates with the successful priming and expansion of antigen-specific T cell responses.

> **Change** (lines 303 - 305): “*This further highlights the importance of depleting CD14⁺ monocytes and CD25⁺ regulatory T cells on day 0 to optimize neoantigen-specific T cell priming.*”

Question 12:

The criteria used to define neoantigens as being mutant specific, mutant selective or wild-type cross-reactive are unclear. The reactivity against the CD8 neoepitope TENM3 1243:S>L appears to be mutant specific, as the baseline or reactivity was ~20% and the response to the wild type peptide was only slightly elevated at the highest peptide concentration. Responses such as the CD4 response against the ATP2C1 427: E>K CD4 epitope should not be considered when evaluating these responses as they are unlikely to recognize any naturally processed and presented antigen.

We appreciate the reviewer’s comments and have clarified the criteria used to categorize neoantigen responses as mutant-specific, mutant-selective, or wild-type cross-reactive. These categorizations were based on pre-defined statistical criteria, which are explicitly included below, in the Methods section (line 834-839), the Results section (215-219 and 315-319) and in the caption of Suppl. Fig. 7 (formerly 6), lines S210-213.

- **Mutant-specific (MT-specific):** Responses were classified as mutant-specific when T cell clones demonstrated significant reactivity to the mutant peptide compared to the unloaded dendritic cell (DC) control and showed no reactivity to the wild-type peptide.

- **Mutant-selective (MT-selective):** Clones were considered mutant-selective if they exhibited significant reactivity to the mutant peptide (over the unloaded DC control), but also displayed some reactivity to the wild-type peptide compared to the unloaded DC control
- **Wild-type cross-reactive (WT cross-reactive):** These responses were defined by significant reactivity to both mutant and wild-type peptides, with no significant difference in magnitude between responses to the two peptides.

TENM3 1243:S>L :

A slight, but statistically significant increase at 3.2 μ M in response to the wild-type peptide compared to the observed baseline reactivity assigns this response to the MT-selective category.

ATP2C1 427:E>K:

The observation that a significant difference was observed between 3.2 μ M and baseline in response to both the mutant and the wildtype peptides, but no difference between mutant and wildtype peptide at 3.2 μ M results in categorizing this response as wild type cross-reactive.

Question 13:

An explanation is needed for the disparities between the relatively high frequencies of some of the CD8 reactivities shown in Fig. 2 and Fig. S5 and the data shown in Fig. 4 where all the frequencies of all of the reactivities, with the exception of 1, presumably that directed against the SRSF1 33:E>K neoepitope, are below 2%.

The disparities in CD8 reactivity frequencies between Fig. 2/S5 and Fig. 4 are primarily due to differences in the number of restimulations. Research-scale experiments included two restimulations, while therapeutic-scale manufacturing involved only one (except for HD17). This reduction in restimulation was a deliberate choice to simplify the process and minimize PRDM1 expression, which negatively correlates with ACT persistence²⁰. Despite lower frequencies, the therapeutic-scale data align with our Nature Medicine dataset¹³, supporting consistency across development phases.

The data presented in Fig. 4C also needs further explanation. Why are all of the points for CD4 reactivities filled but only 1 of the CD8 specificities filled? Do the unfilled points represent the final concentration of these reactivities in the pooled product, and why are some reactivities shown as being in the T cell product before pooling but not in the final product, as all of them should have been present unless they were not pooled?

Fig. 4C shows the detected antigen specific T cell responses (CD8 top, CD4 bottom) for each run at therapeutic scale. The identification and quantification of the CD4 T cell responses were only performed at the vessel level prior to pooling (filled circle) and not at the pooled final drug product (unfilled circle), this has been included in the legend (lines 1103-1104). For CD8 responses the filled circles are prior to pooling and the open circles are from the DP. All vessels contributed to the pooled final drug product, but the concentration for a given response could be lower in the pooled product due to the dilution effect at pooling. In the original Figure we had only shown CD8⁺ responses at the vessel level, for cases where there was not a response to report at the DP level. We understand how this can become confusing, and changed the

presentation to showing all responses, both at the vessel level and at the DP level (see also below).

An explanation is also needed for the data in the supplementary table marked ‘Therapeutic scale’ (should this be labeled Supplementary Table 2?). What is meant by ‘vessel’, or ‘well’, and why are the frequencies sometimes higher in the vessel than in the DP whereas sometimes the reverse is true? How was the final DP generated – were all of the wells containing neoantigen reactive T cells combined, and were there significant differences between the expansions carried out using the different peptide pools? Since multiple wells containing the same peptide pools were set up, were all of the wells pooled, or only the wells that showed effective expansion of reactive T cells pooled, and if so what criteria were used?

At research scale NEO-STIM was performed by inducing multiple replicate wells with an identical peptide set to assess reproducibility, data on each individual well is shared in Suppl. Data S1, first tab of excel file ‘Research scale’. No pooling was performed at research scale.

At therapeutic scale (Suppl. Data S1, second tab of excel file ‘Therapeutic scale’), six parallel vessels with unique peptide sets were used for the manufacturing process. Pooling these six vessels generated the final drug product. The growth kinetics in each vessel vary. As a consequence of that, responses do not get exactly diluted 1:6. Additionally, some vessels are stimulated with related peptides (e.g. a 9mer and a 10mer derived from the same mutation), raising T cell responses that can recognize both peptides. In these instances the frequency in the final DP could unexpectedly be higher, as for example is the case for ENG02 TENM3 1243:S>L (see updated Fig. 4C and Suppl. Data S1, second tab ‘Therapeutic scale’).

> Changes:

- Fig. 4C has been updated to separate out pooled vs pre-pooled frequencies for the CD8 T cell responses.

- The caption for Fig. 4C has been updated to reflect that CD4 responses were only analyzed prior to pooling (lines 1103-1104).

- Lines 308 – 312 have been updated to now include an explanation for the observed variation: *“Interestingly, variations in growth kinetics across vessels and stimulation with related peptides (e.g., 9mer and 10mer from the same mutation) can lead to non-uniform dilution and, in some cases, unexpectedly varying frequencies of antigen-specific T cells in the final drug product, as observed for ENG02 TENM3 1243:S>L (Fig. 4C, Supplementary Fig. 5B and Supplementary Data S1).”*

Question 14:

In general, did de novo/pre-existing reactivity correspond to specificity categories of mut specific/selective, wt-reactive? For DYNC2H1 or RNF103 reactivities, were multiple TCR clonotypes observed in the scRNA data (Fig 5C/5D)? If so, were there distinctions in phenotypic state distribution within or among clusters and did these differ by de novo or pre-existing categories?

We did not see a correlation between de novo vs pre-existing and mutant reactive vs wild-type reactive categories. Since both the number of pre-existing responses as also the number of wild-type reactive clones was quite low, it would be hard to establish a correspondence.

Multiple TCR clonotypes were detected for DYNC2H1 and RNF103, but the small number of cells limited detailed analysis of phenotypic state distribution within or among clusters. The data showed relatively consistent distribution (UMAPs, first and third plots), and for clarity, we collapsed the data in the manuscript (UMAPs, first and second plot) to focus on high-level conclusions about the transcriptional profiles of these cells.

Reviewer 2 feedback:

This manuscript describes the development of NEO-STIM, a platform that induces de novo and expands memory T cell responses to patient-specific neoantigens ex vivo from peripheral blood. The authors convincingly demonstrate that the resulting drug products (DPs) are polyclonal, polyfunctional, mutant-selective, and cytotoxic, with clear recognition of autologous tumor. Their platform overcomes several limitations of TIL-based ACT, including reliance on resectable tumors and exhausted phenotypes, and is supported by preclinical and early clinical feasibility data.

All major conclusions are well supported, and the methodological detail is commendable. However, several revisions would improve clarity and reproducibility:

Major Concerns:

Question 1:

Please explicitly state which subsets are depleted (e.g., CD14⁺ monocytes, CD25⁺ Tregs), whether CD11c⁺ DCs are enriched, and what maturation cocktail is used. Consider including a side-by-side comparison of DC phenotypes and costimulatory marker expression after maturation to validate the superiority of naturally circulating DCs over moDCs.

We appreciate the reviewer's request for clarification. CD14⁺ monocytes and CD25⁺ Tregs are depleted on Day 0, as specified in the manuscript (Results: lines 125-127; Methods: line 712; all line references refer to the updated clean version). CD11c⁺ DCs are utilized at their natural abundance following depletion, without further enrichment. This is now clarified in the updated Figure caption of Fig. 1B, as well as in the Methods (lines 713-714: "For research experiments where additional populations were depleted, ...". The maturation cocktail components and procedures are detailed in the Methods section (lines 720-723) and have also been added to updated Figure caption 1B.

To address the reviewer's suggestion, we measured DC costimulatory markers for conventional DC subsets (cDC1 and cDC2) and moDCs. A direct side by side comparison of these is difficult, as the protocols differ. However, as expected, moDCs show high expression of CD40, CD80, CD83 and CD86 two days post maturation. Likewise, naturally occurring cDCs show robust costimulatory marker expression two days post maturation.

We have included these data in the manuscript as Suppl. Fig. 3B, see below.

Finally, the primary motivation to investigate naturally occurring DCs (nDCs) over moDCs was to streamline and shorten the manufacturing protocol, rather than identifying a superior subset. In the first iteration of the protocol, PBMCs were depleted of CD14⁺ cells for the reasons indicated in the manuscript (removal of potentially inhibitory cells), and the CD14⁺ cell population was used to prepare moDCs, which were used to facilitate T cell priming and expansion. We hoped to avoid this separate workstream by using nDCs. Interestingly, most parameters tested were comparable or superior which led us to continue with nDCs. The robust expression of costimulatory markers on cDCs is in line with this decision. We tried to avoid general claims of nDC superiority by stating in several locations that nDC preference is seen as part of streamlining NEO-STIM (lines 120, 128-129, 160, 162).

>Changes:

Add statement that CD11c⁺ cells stem from the CD14/CD25 depleted cell pool, and details on maturation cocktail to caption of Fig. 1B (lines 1000-1003): “25-50 million depleted cells, containing CD11c⁺ naturally circulating dendritic cells, were cultured overnight in the presence of FMS-like tyrosine kinase 3-ligand (FLT3L), then peptide loaded, matured with maturation cocktail (TNF- α , interleukin (IL)-1 β , Prostaglandin-E 1 and IL-7), and cultured for 14 days with IL-7 and IL-15.”

Clarify in Methods, that depletion of CD11b⁺ or CD11c⁺ cells was only performed for dedicated research experiments (lines 713-714).

Question 2:

The magnitude of neoantigen-specific responses varies widely across patients. If feasible, correlate DP response magnitude with pre-culture CD8⁺ T cell frequency or baseline TCR repertoire richness. At minimum, please discuss whether predictive biomarkers (e.g., mutation burden, CD8⁺/CD4⁺ ratio, memory subset composition) might inform response yield and product quality.

We appreciate the reviewer’s suggestion regarding predictive biomarkers. In our current dataset, CD8⁺ T cell frequency positively correlates with neoantigen-specific response yield, as shown by the hit rate metric (Fig. 4B). This highlights CD8⁺ abundance as a potential predictor of response yield.

As our dataset grows, we will continue to evaluate additional biomarkers, such as mutation burden, CD8⁺/CD4⁺ ratio, memory subset composition, and TCR repertoire richness, to refine inclusion criteria and optimize manufacturing decisions. Similarly, in the context of our recent publication in Nature Medicine¹³ we performed an analysis to try and correlate various characteristics (as mentioned by the reviewer) with clinical outcome. We did not find any predictors here but notably observed tumor reduction in patients treated with very comparable drug products and sometimes low neo-antigen specific T cell content. Identifying robust biomarkers remains a key priority to enhance product quality and therapeutic outcomes.

Question 3:

The authors report a minority of responses with WT reactivity. Please summarize the percentage of tested responses falling into each specificity category (mutant-specific, selective, WT-crossreactive). If possible, add data on functional avidity (e.g., EC50 curves) for at least one WT-

crossreactive versus mutant-specific clone to support safe discrimination. Clarify whether WT-reactive clones would be excluded from clinical release criteria.

We thank the reviewer for this question.

In our dataset, most T cell responses were mutant-specific (7/19; 36.8%) or mutant-selective (9/19; 47.4%), with a minority showing WT cross-reactivity (3/19; 15.8%). This is illustrated in Suppl. Fig 7 (formerly Suppl. Fig. 6) and summarized at the high level in lines 319-321.

Functional avidity data were generated for four TCRs (SRSF1: n=3, one TCR with 2 clonotypes; ARAP1: n=1), and showed mutant-specific binding with avidities in the nano- to micromolar range (New Suppl. Fig. 8).

In the clinical setting, we similarly observed predominantly tumor-specific responses with comparable EC_{50} values (Borgers *et al.*¹³, Extended Data Fig. 4). Given the infrequent observation of WT-reactive responses during development, the clinical approach involved infusing bulk products and assessing cross-reactivity post-infusion. Importantly, no safety signals directly attributable to neoantigen-specific products were observed in treated patients¹³.

> **Change:**

A new Suppl: Fig. 8 has been created which shows the generated EC_{50} data for four TCRs derived from this study. Lines 251 – 252 of the manuscript have been updated to reflect this.

Question 4:

PRDM1 induction after repeated stimulation suggests terminal effector skewing. The authors might consider comparing $CD62L^+CCR7^+$ memory phenotypes across $1\times$ vs $2\times$ stimulation DPs to assess whether reduced stimulation could preserve less-differentiated states. A brief analysis of exhaustion marker expression (e.g., PD-1, LAG-3) in the final product would also be informative.

We thank the reviewer for this suggestion and have performed additional analysis.

In conjunction with the upregulation of *PRDM1* over time, we also show that the ‘partial effector cells’ (low/absent levels of *CCL3*, *IFNG* and *TNFRSF9*) express exhaustion markers (*PDCD1*, *TIGIT* and *TOX*) (Suppl. Fig. 9E, formerly Suppl Fig. 7E).

Additionally, as suggested by the reviewer, we examined the expression of stem-like (*CCR7*, *SELL*) and exhaustion markers (*LAG3*, *PD1CD1* & *PRDM1*) at different times of the protocol. Little expression (no more than in bystander cells) was observed for *CCR7* and *SELL*. *LAG3* and *PRDM1* both increased expression over time, with stable *PDCD1* expression. This pattern was consistent across both the SRSF1 and ARAP1 T cell responses. These findings support the interpretation that repeated stimulation drives terminal effector differentiation and support our decision to manufacture a product with only two stimulations to preserve a drug product with a less differentiated state.

G

> Change:

The above data has been added as panel G to Suppl. Fig. 9, with the following text added to the manuscript (lines 281-285): “We analyzed stem-like (*CCR7*, *SELL*) and exhaustion markers (*LAG3*, *PDCD1*, *PRDM1*) in both responses (Supplementary Fig. 9G). *CCR7* and *SELL* showed minimal expression at all timepoints, while *LAG3* and *PRDM1* increased over time with stable *PDCD1* levels. This pattern was consistent across SRSF1 and ARAP1 T cell responses.”

Question 5:

In the patient with tracked TCR clonotypes, the activated/exhausted phenotype is well described. If possible, include a quantitative comparison of persistence or expansion between pre-existing and de novo clonotypes. This could clarify the relative durability of newly induced versus expanded memory clones.

Across this manuscript and the clinical study¹³ quantitative analyses indicate that the more robust persistence and expansion following infusion is associated with pre-existing clonotypes. This is likely explained due to their higher abundance within the infused product, rather than an intrinsic phenotypic advantage, given the phenotypic similarity between those two groups:

- This manuscript (lines 344-346): “Across two responses which initially were considered pre-existing, three out of eight subclones were de novo. One of those three was detected post-infusion, compared to five out of five which were pre-existing (Figure 5A, B).”

- Borgers et al.¹³ (Fig. 5C, D): Consistent results were observed in the clinical context. Two out of six clonotypes from a pre-existing response appeared to be de novo at the clonotype level. Neither was detected in blood or tumor post-treatment, compared to three out of four pre-existing clonotypes.
- Additionally, in Borgers et al.¹³, Ext. Fig 2C, D, we found that both pre-existing and de novo clonotypes exhibited a diverse array of phenotypes, with no clear phenotypic distinction between the two groups. However, pre-existing responses were more abundant in the drug product, both by frequency and by the number of detected clonotypes. This higher abundance within the infused product may explain why de novo responses, despite phenotypic similarity, are not always detected post-treatment and their lower initial abundance may make expansion to detectable levels less likely.

Question 6:

Would this approach work in tumors with fewer mutations (e.g. some epithelial cancers or pediatric cancers)? It likely can, as long as at least a handful of neoantigens are identifiable, but the effort (60 peptides per patient) might yield fewer hits. A sentence discussing this (perhaps noting that even low-mutational-load tumors can still have a few strong neoantigens to target) would be useful to frame the breadth of this therapy.

We appreciate the interest in the application of NEO-STIM in other indications with fewer mutations. We performed NEO-STIM in three additional patients with ovarian cancer (OVC), with a tumor mutational burden (TMB) ranging from 25 – 88. This resulted in 40-57 predicted neo-antigens to raise responses towards. In addition to the patient-specific mutations, an additional set of model antigens was added in one (PT07 and PT08) or two (PT09) pools for each patient in case fewer hits were observed, as suggested by the reviewer. Consistent with the results obtained in PBMCs derived from melanoma patients, these studies demonstrate that NEO-STIM at research scale:

- Generates polyclonal drug products in patients with low TMB that recognize patient-specific mutations as well as model antigens:
2-13 CD8⁺ T cell responses and 1-13 CD4⁺ T cell responses (*n*=3 donors)
- Induced neoantigen-specific T cell responses produce IFN γ , TNF α and CD107a in response to cognate peptide.

> Changes:

The data derived from NEO-STIM in PBMCs derived from OVC patients is included in the new Suppl. Fig. 6 shown below. Text in the manuscript has been updated to reflect the addition, and response numbers were updated in the two Results sections starting in line 160 and 200 (see also Redlines of the manuscript).

New Suppl. Fig. 6 illustrating results from NEO-STIM with materials derived from Ovarian carcinoma patients.

Question 7:

Additionally, while the focus is lab/process development, the authors did treat at least one patient and mention an ongoing Phase I trial. If possible, the clinical context should be expanded slightly: For example, the authors state BNT221 showed early signals of efficacy – without divulging too much beyond what is allowed, can they say what signals? (Tumor regression in some patients? Durable T cell persistence?) Even a general statement like “early signs of anti-tumor activity were observed in the trial, such as objective tumor responses in some participants” would underline the importance. If such details are not yet public, it’s understandable, but then perhaps emphasize that a formal clinical trial is underway to assess safety/efficacy, which is a strength of this work’s translational readiness.

We appreciate the reviewer request on sharing some safety and efficacy data of BNT221 to further strengthen the work’s translational readiness. As reported in Borgers et al.¹³, the early safety and efficacy data of BNT221 in heavily pretreated patients are encouraging. We have updated the manuscript to briefly describe the observed signals and refer to the reported data to underscore the translational aspect of BNT221¹³.

> Change:

We have updated the Discussion to include additional detail from the clinical evaluation (lines 432 - 434: “Of the nine treated patients, six had stable disease (SD) as their best overall response and four of these patients with SD showed tumor regression after infusion. The drug product characteristics observed in the FIH study are consistent with the findings described here.”

Question 8:

I recommend adding a sentence on how this approach might integrate with other therapies – for instance, will patients receive checkpoint inhibitors alongside NEO-STIM T cells? This could be relevant since ex vivo expanded cells might still benefit from checkpoint blockade post-transfer. Even if not part of this study, acknowledging combination strategies (as is common with TIL therapy + IL-2 or checkpoints) would show awareness of the clinical landscape.

We appreciate the reviewer’s comment regarding the potential integration of NEO-STIM-generated drug products with combination therapies, particularly checkpoint inhibitors. Indeed, translational data from the FIH study demonstrated robust PD-1 upregulation on multiple post-infusion neoantigen-specific T cell responses (Borgers et al.¹³, Fig. 4D), highlighting a compelling rationale for PD-1 blockade as a complementary strategy. This observation aligns with the broader clinical landscape, where checkpoint inhibition has proven synergistic with adoptive cell therapies, further underscoring the potential of NEO-STIM in combination approaches to enhance therapeutic efficacy.

> Change:

We added this perspective to the Discussion (lines 434-436): *“Furthermore, evaluation of combination strategies with for example checkpoint inhibitors and/or cytokines is warranted to enhance the potency, expansion and persistence of neoantigen specific cells further.”*

Minor Concerns:

Question 9:

It’s excellent that each peptide pool was run in triplicate to test process reproducibility. The supplemental data likely show high correlation (the text notes $R^2=0.98$ between some replicates). The authors might consider mentioning in the Results that reproducibility between parallel cultures was high, as this will give readers confidence that the process yields consistent results. A brief mention of the replicate consistency (perhaps referencing Supplementary Fig. S5 or S1B as appropriate) would suffice.

We appreciate the reviewers’ suggestion to strengthen the paper and have incorporate this in the Results.

> **Change** (lines 188-190): *“Reproducibility between cultures was consistent across biological replicates, in particular for higher frequency responses (Supplementary Data S1).”*

Question 10:

In the main figures, some data (e.g., Fig 2A examples, 2H cytotoxicity, 2I tumor recognition) seem to highlight one or two patients. It would be helpful if the authors could ensure that data from all four patients are reported, at least in aggregate. For example, Fig. 2C, 2E summarize the number of responses per patient – presumably each dot is one patient’s result. Clarify in the figure caption if those are per-patient medians or individual responses. Similarly, for cytotoxicity and tumor recognition, if not all patients were tested, state the number of patients or responses

evaluated. Transparency about n for each assay will help readers gauge the consistency across the cohort.

We appreciate the reviewer's request for transparent reporting.

Fig 2A shows 1-2 examples for CD8⁺ tetramer staining per patient and tetramer staining graphs for all tested responses are shown in Suppl. Fig. 5. We have added a reference to Suppl. Fig. 5 in the caption of Fig. 2A.

Fig. 2B and D show two examples each to illustrate the method, and the respective aggregate data is shown in Fig 2C, E. This is explained in the Figure caption. Groups of responses are shown per patient as defined in the Figure caption. Datapoints show all responses per patient, as listed in the caption Fig. 2C, and this is now also added in the caption of Fig. 2E.

For Fig. 2F-I we now clarified in the Figure caption that only responses with sufficient sample availability and magnitude of response were tested.

Fig. 2F shows representative examples as listed in the caption; the total n is given in the caption and in the main text (line 205).

Fig. 2G shows example graphs as specified in the caption and we added a reference in the caption to Supplementary Fig. 7, which shows all responses. Results on all samples tested are given in the main text (lines 214-218).

Fig 2H shows all responses that were positive, as described in the text (line 229), where also the total number of responses tested is given.

For Figure 2I we explain in the text, why only one patient was investigated at this point (lines 234-235): "*We were able to obtain a single-cell suspension from a tumor of one patient (PT04).*"

Figure 4D shows examples, for which a high-level summary is given in the table below the graphs, and graphs for all specificity plots are provided in Suppl. Fig. 7 (formerly 6). We added a note to refer the reader to Suppl. Fig. 7 in the Figure caption of Fig. 4D (line 1111). Total sample number and number of specific responses is also given in the main text (lines 314-315; 319-321).

For Fig 4E, we added the sample availability limitation in the main text (line 325). This consideration is also reflected in the Reporting Summary under the section "Sample size" (Header "Life Sciences Study Design").

Suppl Figure 10C (formerly 8C) shows representative examples for 26 samples tested; we added the word "representative" to the Figure caption for clarity. We clarified the overall result in the Figure caption and in the main text (line 313).

Suppl Fig. 10D, E states in the Fig caption (lines S265-2669: "*Six responses tested and found cytotoxic, of which five are shown here.*"

> Changes:

Added information to Figure legends of Fig 2A, 2E, 2F-I, 2G, 4D, Suppl. Fig. 10C and to main text (lines 313, 325).

Question 11:

Overall, the figures are well-designed. A few suggestions:

(i) Figure 1B – if this is a schematic of the process, ensure that all steps are labeled clearly (e.g., “Deplete PBMC subsets,” “Primary stimulation (Stim1) day 0–14,” etc.). Currently it’s referenced that Fig. 1B shows process steps; just make sure it’s readable and perhaps include an explanation in the legend for any icons or abbreviations.

Figure 1B has been updated as suggested.

>Change: Update Figure 1B and its caption.

(ii) Figure 2 – panels C, E (showing range and medians of responses) should indicate whether the percentages shown are fraction of total CD8 or CD4 T cells that are neoantigen-specific (I assume yes). Adding “% of CD8⁺ T cells” in the axis label or caption would help. Also, define “tet⁺” in the caption for non-specialist readers (T cells detected by peptide-MHC tetramers).

The definition of tet⁺ has been included in the legend of this figure at first use. The suggested label for the axis / caption was used.

>Change: Figure caption 2A

(iii) Figure 3 – the UMAP plots and heatmaps from scRNA-seq are complex. The authors might add a brief description in the legend about what the major clusters represent (activated effector vs memory, etc., as derived from Fig. S7). A simplified summary of the key takeaway (that ARAP1 de novo clone shifts to effector state early, vs the SRSF1 memory clone which shifts gradually) could guide the reader through panel E/F.

We thank the reviewer for this recommendation and have included the following in the caption of Fig. 3 E-F (lines 1071-1075): “*Tem subsets are characterized by expression of IL7R, effector subsets by expression of TNFRSF9, and partial effectors by low levels of TNFRSF9 combined with exhaustion markers (see Supplementary Fig. 9). Note the larger phenotypic shifts in response to antigen for the ARAP1 de novo clonotypes than the SRSF1 memory clonotypes, especially at the first stimulation (STIM1) timepoint.*”

>Change: Add explanation of UMAP results in figure caption 3E-F (lines 1071-1075)

Question 12:

Please ensure all abbreviations are defined on first use. For instance, “DP” (drug product) is defined in the Introduction – that’s good. Also define Stim1/2/3 clearly as rounds of stimulation. The term “research scale” vs “therapeutic scale” is used; perhaps clarify that “research scale” refers to small-scale lab process used for initial 4 patients, and “therapeutic scale” is a full-scale manufacture (if I understood correctly). Minor point: “WT” and “MT” (wild-type vs mutant) should be defined in figure legends or methods for clarity.

We have incorporated these suggestions and added explanations for the acronyms shown below.

WT and MT is explained in Figure captions Fig. 2 and 4, Suppl. Fig. 7 (formerly 6) at first use (i.e. not in every subpanel), and we have spelled it out in Suppl. Fig. 10 (formerly 8).

Research and Therapeutic scale is best explained based on the size of culture vessels, and this has been clarified in the Methods (lines 731-733): “*Plating was either as 2 mL in a 24-well plate in triplicates (research scale) or as 9 mL in a 6-well plate or 10M or 100 M G-Rex vessel (therapeutic scale; Wilson Wolf),...*”

> Changes:

Spell out additional abbreviations at first use: TAA (line 47), TME (line 73), PBMCs (line 84), HLA (line 90), GMP (lines 110-111), MHC (line 167-168), pre-NEO-STIM and STIM 1,2,3 (line 245-247); wild-type and mutant spelled out in caption of Suppl. Fig. 10 (formerly 8); research scale vs therapeutic scale clarified in Methods (lines 731-733)

Question 13:

The authors cite relevant literature on TIL therapy and neoantigen vaccines. One suggestion is to cite recent clinical successes: for instance, the FDA approval of Lifileucel for melanoma in 2024, which they allude to in the introduction, could be explicitly referenced as a milestone that underscores the importance of advancing neoantigen-targeted T cell therapies. Additionally, if there are any comparable approaches (perhaps a recent study attempting to prime T cells from blood for adoptive therapy, if any), mentioning those would position this work in the landscape. It appears this approach is quite unique, but for completeness, the authors might consider if groups have tried similar “prime-and-expand” T cell therapies (I recall some early trials expanding T cells to tumor-associated antigens or virus antigens ex vivo). A sentence distinguishing NEO-STIM from TCR-engineered cell therapies would also help: e.g., emphasize this does not require genetic modification, making it potentially faster and less expensive once neoantigen peptides are identified.

We appreciate the reviewers’ suggestions to strengthen the manuscript and have adopted this to enhance the positioning of our work in the landscape.

> Changes:

Added the following statement (line 46-47): “*...resulting in FDA approval of a tumor-infiltrating lymphocyte (TIL) product, defining a major milestone for personalized adoptive T cells therapies.*”

Added reference (Khateb et al.²¹) to line 48-49, and expanded on the landscape

Reviewer 3 feedback:

This study develops and validates NEO-STIM, an ex vivo method to generate personalized neoantigen-specific T-cell responses under optimized conditions that avoid immunosuppression. Lenkala et al. validated the protocol using model antigens with healthy donor PBMCs expressing relevant HLA types. They enhanced T cell priming by depleting inhibitory cell subsets and identified the best APC populations by assessing expansion, diversity, and functionality of CD8+ responses—including those to long peptides that usually induce CD4+ T cells. Applying the method to PBMCs from four melanoma patients, they detected both pre-existing and newly induced CD8+ responses, via combinatorial pMHC tetramer staining – as well as CD4+ T cell responses. The generated DP T cells were further phenotypically and functionally characterized. Finally, the protocol was optimized for therapeutic-scale manufacturing while maintaining functional activity.

The study addresses an important and unmet clinical need with a promising approach. The idea is compelling and has strong potential for personalized immunotherapy. However, the work feels preliminary and would benefit from more extensive analysis and deeper characterization to strengthen its conclusions and clinical relevance.

Major concerns:

Question 1:

Limited donor and patient diversity: Proof-of-concept experiments use only two healthy donors (one with weak de novo reactivity), with very low response frequencies (<1%), raising questions about functional relevance. The clinical cohort is restricted to melanoma patients, nearly all previously treated with immunotherapy, limiting generalizability and potentially confounding immune responses.

We agree with the reviewer that donor and patient diversity is critical for demonstrating the robustness and generalizability of NEO-STIM. The platform development and therapeutic validation included data from 22 healthy donors and six melanoma patients, as presented across the main and supplementary figures. Additionally, we extended NEO-STIM to three ovarian cancer patients with low tumor mutational burden (TMB 25–88), generating polyclonal drug products with multiple effector functions, consistent with results from melanoma-derived PBMCs. Furthermore, a recent publication from our group describing a modified NEO-STIM protocol, optimized for mutant KRAS¹⁶, demonstrated reproducible induction of T cell responses against this shared neoantigen across 47 KRAS-MHC combinations in healthy donor PBMCs, underscoring the applicability of NEO-STIM beyond melanoma and its broader translational potential.

> Changes:

The data derived from NEO-STIM in PBMCs derived from OVC patients is included in the new Suppl. Fig. 6. Text in the manuscript has been updated to reflect the addition, and response numbers were updated in the two Results sections starting in line 160 and 200 (all line references refer to the updated clean version). See also Redlines of the manuscript for updates of numbers.

Question 2:

Lack of tumor antigen expression and clonality data: The manuscript does not report expression levels or clonal distribution of the selected neoantigens on patient tumors, which is critical for evaluating clinical impact.

We agree that this data would be informative. We have now included this data in Supplemental Data S1, i.e. mutant allele fractions (column P 'variant_dna_allele_fraction') and mutant-allele-specific RNA-Seq expression (in TPM, column O 'variant_rna_expression_tpm') on tabs "Research scale" and "Therapeutic scale". In some cases, model antigens (neoantigens or TAAs) were additionally used in peptide pools^{6-8,22-24}. Given that these antigens were not part of the patient's own mutational landscape, mutant allele fraction and mutant-allele-specific RNA-Seq expression were marked as N/A in those cases.

>Changes: Addition of mutant allele fractions and mutant-allele-specific RNA-Seq expression in Supplemental Data S1 (columns P and O)

Question 3:

Insufficient functional validation and controls: Functional assays with autologous tumor cells are limited (only two responses tested), and killing assays are lacking. Healthy autologous cells were not used as specificity controls. Many experiments rely on single replicates, and controls often lack APC plus non-mutated peptide conditions, weakening robustness and specificity assessment.

We respectfully disagree with the reviewer's assessment regarding the robustness and specificity of our functional validation. Autologous tumor material was indeed limited, but we prioritized this for demonstration of tumor recognition in a physiologically relevant setting, evaluating three T cell responses across two patients (Fig. 2I and 4E, $n=3$ technical replicates).

During development each peptide pool was run in triplicate (and for Ovarian carcinoma in quadruplicate) to test process reproducibility. Data is provided in Suppl. Data S1 and the high reproducibility is now specifically called out in the main text.

Cytotoxic capacity was determined by quantifying the fraction of active caspase 3 positivity against A375 tumor lines transduced with the HLA restriction element and a construct encoding 200 amino acids of mutant or wildtype sequence, such that processing and presentation must occur for tumor recognition and killing. Evaluation of the mutant and wildtype (i.e. non-mutated, representative of healthy cells) constructs, and assessment against the parental line with expression of the relevant HLA only, allowed us to assess specificity (Fig. 2H, $n=3$ technical replicates).

Finally, specificity was also confirmed through a co-culture assay, where APCs loaded with either the wildtype (i.e. non-mutated) or mutant peptide were co-cultured with the neoantigen specific T cell responses (Fig. 2G and Supplementary Fig. S6, $n=3$ technical replicates). The observed functionality profile showed that 84.2% (16 out of 19 responses tested) preferentially recognized the mutant epitope.

Collectively, these experiments incorporated appropriate controls, technical replicates, and specificity assessments, generating robust and reproducible data that validate the functionality and cytotoxic capacity of the neoantigen-specific T cell responses.

>Change: Added statement on reproducibility in main text (lines 188-190): “*Reproducibility between cultures was consistent across biological replicates, in particular for higher frequency responses (Supplementary Data S1).*”

Question 4:

Use of artificial tumor models: Validation frequently relies on tumor cells transduced to express the desired neoantigens, which may not reflect physiological expression or clinical relevance. Confirmation with authentic autologous tumor cells is necessary.

We agree with the reviewer that autologous tumor cells represent the most physiologically relevant target for validation in killing assays. However, the availability of such material is inherently challenging, particularly in the context of a non-clinical study. For this manuscript, we leveraged autologous tumor material from two patients to evaluate three T cell responses (Fig. 2I and 4E, n=3 technical replicates), but no additional material was accessible.

Importantly, the clinical efficacy data for BNT221 shown in Borgers *et al.* further supports the preclinical findings described here. In the FIH study, neoantigen-specific drug products manufactured according to the process outlined in this manuscript demonstrated tumor reductions post-infusion in heavily pre-treated patients, accompanied by infiltration of neoantigen-specific T cells into the tumor. These clinical observations substantiate the relevance and translational potential of our approach¹³.

Question 5:

Incomplete phenotypic and CD4+ T cell characterization: Phenotypic analysis is largely restricted to CD8+ T cells and compares only one pre-existing and one de novo response. There is no thorough characterization of other CD8+ or CD4+ T cell responses. Additionally, no pre-protocol assessment of CD4+ responses was conducted to distinguish pre-existing from induced de novo populations, limiting interpretation of efficacy.

CD4+ T cell responses were characterized using recall response assays, which included pre-NEO-STIM PBMCs as a baseline control (this is now specifically included in the Methods, line 819). No CD4+ responses were observed at baseline, supporting the interpretation that these responses were induced de novo. As noted in the discussion (line 379), the recall assay does not achieve the same specificity threshold as pMHC multimers used for CD8+ T cells, which may partially account for the absence of detectable pre-existing CD4+ responses. Unfortunately, the lack of suitable multimer reagents for CD4+ T cells limited our ability to perform sequence-based analyses for deeper characterization. Despite these technical constraints, the recall assay provided meaningful insights into the induction of CD4+ T cell responses through NEO-STIM.

> Changes: to enhance clarity, we specified the use of pre - NEO-STIM PBMCs in the Methods, line 819: “*Pre - NEO-STIM PBMCs and induced T cells...*”

Minor concerns:

Question 6:

Proof-of-concept responses should be shown separately for high and low immunogenic antigens to support claims on low immunogenic antigen recognition.

We understand the reviewer's request to categorize antigens into high and low immunogenicity groups, as was done for several Supplementary Figures using model antigens (Suppl Fig 1E, Suppl 3F).

However, for the therapeutic-scale experiments performed with patient samples in this manuscript, private patient-specific neoantigens were used, and by-and-large, no prior data on their immunogenicity was available, reflecting the clinical context. As such, classification of these antigens into high or low immunogenicity categories was not feasible within the experimental framework. This approach mirrors real-world scenarios, where neoantigen immunogenicity is often unknown prior to therapeutic application. For a subset of patients (PT3, PT7, PT8 and PT9) 1-2 pools of previously identified neo-antigens were included^{6-8,22-24}, Supplemental Data Set1 displays whether or not responses were raised towards those for each peptide.

Question 7:

Activation markers should be shown separately for clarity.

We have taken the reviewers' suggestion and have grouped the expression profile in various categories, including activation markers.

> **Change:** Add labels for activation and other markers in Fig. 5C, D

Question 8:

Supplementary Figure S1E is hard to interpret and uses different donors than the previous experiments, making it difficult to get an overall picture.

During the development of NEO-STIM materials derived from a total of 22 healthy donors and six melanoma patients were used. While certain parts of the process (i.e. priming vs expansion) have been optimized with a different set of healthy donors, each experiment was internally controlled. The final proof of concept experiments were run through the same process across multiple patient samples.

References:

- 1 Kristensen, N. P. *et al.* Neoantigen-reactive CD8+ T cells affect clinical outcome of adoptive cell therapy with tumor-infiltrating lymphocytes in melanoma. *Journal of Clinical Investigation* **132** (2022). <https://doi.org/10.1172/jci150535>
- 2 Linnemann, C. *et al.* High-throughput epitope discovery reveals frequent recognition of neo-antigens by CD4+ T cells in human melanoma. *Nat Med* **21**, 81-85 (2015). <https://doi.org/10.1038/nm.3773>
- 3 Parkhurst, M. R. *et al.* Unique Neoantigens Arise from Somatic Mutations in Patients with Gastrointestinal Cancers. *Cancer Discov* **9**, 1022-1035 (2019). <https://doi.org/10.1158/2159-8290.CD-18-1494>
- 4 Robbins, P. F. *et al.* Mining exomic sequencing data to identify mutated antigens recognized by adoptively transferred tumor-reactive T cells. *Nat Med* **19**, 747-752 (2013). <https://doi.org/10.1038/nm.3161>
- 5 Tran, E. *et al.* Immunogenicity of somatic mutations in human gastrointestinal cancers. *Science* **350**, 1387-1390 (2015). <https://doi.org/10.1126/science.aad1253>
- 6 Fritsch EF, R. M., Ott PA, Brusica V, Hacohen N, Wu CJ. HLA-binding properties of tumor neoepitopes in humans - PubMed. *Cancer immunology research* **2** (2014). <https://doi.org/10.1158/2326-6066.CIR-13-0227>
- 7 Pittet, M. J. *et al.* High Frequencies of Naive Melan-a/Mart-1-Specific Cd8+ T Cells in a Large Proportion of Human Histocompatibility Leukocyte Antigen (Hla)-A2 Individuals. *The Journal of Experimental Medicine* **190**, 705-716 (1999). <https://doi.org/10.1084/jem.190.5.705>
- 8 van Buuren MM, C. J., Schumacher TN. High sensitivity of cancer exome-based CD8 T cell neo-antigen identification - PubMed. *Oncoimmunology* **3** (2014). <https://doi.org/10.4161/onci.28836>
- 9 Collin, M. & Bigley, V. Human dendritic cell subsets: an update. *Immunology* **154**, 3-20 (2018). <https://doi.org/10.1111/imm.12888>
- 10 F, M. & T, L. Comprehensive Phenotyping of Human Dendritic Cells and Monocytes - PubMed. *Cytometry. Part A : the journal of the International Society for Analytical Cytology* **99** (2021 Mar). <https://doi.org/10.1002/cyto.a.24269>
- 11 Schlitzer, A. *et al.* Identification of cDC1- and cDC2-committed DC progenitors reveals early lineage priming at the common DC progenitor stage in the bone marrow. *Nat Immunol* **16**, 718-728 (2015). <https://doi.org/10.1038/ni.3200>
- 12 Andersen, R. S. *et al.* Parallel detection of antigen-specific T cell responses by combinatorial encoding of MHC multimers. *Nat Protoc* **7**, 891-902 (2012). <https://doi.org/10.1038/nprot.2012.037>
- 13 Borgers, J. S. W. *et al.* Personalized, autologous neoantigen-specific T cell therapy in metastatic melanoma: a phase 1 trial. *Nature Medicine* (2025). <https://doi.org/10.1038/s41591-024-03418-4>
- 14 de Greef, P. C. *et al.* The naive T-cell receptor repertoire has an extremely broad distribution of clone sizes. *Elife* **9** (2020). <https://doi.org/10.7554/eLife.49900>
- 15 Oakes, T. *et al.* Quantitative Characterization of the T Cell Receptor Repertoire of Naive and Memory Subsets Using an Integrated Experimental and Computational Pipeline Which Is Robust, Economical, and Versatile. *Front Immunol* **8**, 1267 (2017). <https://doi.org/10.3389/fimmu.2017.01267>
- 16 Conn, B. P. *et al.* Generation of T cell responses against broad KRAS hotspot neoantigens for cell therapy or TCR discovery. *Cell Reports Methods* **5**, 101049 (2025). <https://doi.org/10.1016/j.crmeth.2025.101049>

- 17 Lemieux, A. *et al.* Enhanced detection of antigen-specific T cells by a multiplexed AIM assay. *Cell Rep Methods* **4**, 100690 (2024). <https://doi.org/10.1016/j.crmeth.2023.100690>
- 18 Zheng, M. Z. M. *et al.* Deconvoluting TCR-dependent and -independent activation is vital for reliable Ag-specific CD4⁺ T cell characterization by AIM assay. *Science Advances* **11**, eadv3491 (2025). <https://doi.org/doi:10.1126/sciadv.adv3491>
- 19 Vyasamneni, R. *et al.* A universal MHCII technology platform to characterize antigen-specific CD4(+) T cells. *Cell Rep Methods* **3**, 100388 (2023). <https://doi.org/10.1016/j.crmeth.2022.100388>
- 20 Yoshikawa, T. *et al.* Genetic ablation of PRDM1 in antitumor T cells enhances therapeutic efficacy of adoptive immunotherapy. *Blood* **139**, 2156-2172 (2022). <https://doi.org/10.1182/blood.2021012714>
- 21 Khateb, M. *et al.* Rapid enrichment of progenitor exhausted neoantigen-specific CD8 T cells from peripheral blood. *bioRxiv* (2025). <https://doi.org/10.1101/2025.05.11.653315>
- 22 Ott, P. A. *et al.* An immunogenic personal neoantigen vaccine for patients with melanoma. *Nature* **547**, 217-221 (2017). <https://doi.org/10.1038/nature22991>
- 23 Ott, P. A. *et al.* A Phase Ib Trial of Personalized Neoantigen Therapy Plus Anti-PD-1 in Patients with Advanced Melanoma, Non-small Cell Lung Cancer, or Bladder Cancer. *Cell* **183**, 347-362 e324 (2020). <https://doi.org/10.1016/j.cell.2020.08.053>
- 24 Sahin U, D. E., Miller M, Kloke BP, Simon P, Löwer M, Bukur V, Tadmor AD, Luxemburger U, Schrörs B, Omokoko T, Vormehr M, Albrecht C, Paruzynski A, Kuhn AN, Buck J, Heesch S, Schreeb KH, Müller F, Ortseifer I, Vogler I, Godehardt E, Attig S, Rae R, Breitkreuz A, Tolliver C, Suchan M, Martic G, Hohberger A, Sorn P, Diekmann J, Ciesla J, Waksman O, Brück AK, Witt M, Zillgen M, Rothermel A, Kasemann B, Langer D, Bolte S, Diken M, Kreiter S, Nemecek R, Gebhardt C, Grabbe S, Höller C, Utikal J, Huber C, Loquai C, Türeci Ö. Personalized RNA mutanome vaccines mobilize poly-specific therapeutic immunity against cancer - PubMed. *Nature* **547** (2017). <https://doi.org/10.1038/nature23003>

Reviewer #1 (Remarks to the Author):

The majority of issues raised by the reviewers were addressed in the revised manuscript; however, there were a few additional points that need to be clarified.

In response to question 5 from reviewer 1, the distinction between highly immunogenic and weakly immunogenic still seems artificial. Highly immunogenic antigens might simply possess a larger repertoire of reactive clonotypes and therefore might elicit a more robust response. While this is of some interest, the most important point regarding tumor antigens is their relative potency against natural tumor antigen targets, as this can vary widely depending on the target.

We agree with the reviewer that the size of the TCR repertoire can play a role. We have included this as a possible explanation in the text. Additionally, we are referencing the section using patient specific targets as a point of value to assess relative potency against natural tumor antigen targets.

> Changes:

- lines 86 – 88: *“The size of the TCR repertoire and epitope characteristics are key aspects that impact the generation of a T cell response.”*

- lines 93 – 95: *“Finally, T cell products were generated from patient derived PBMCs and their relative potency was measured against engineered and natural tumor targets.”*

In response to question 12 from reviewer 1, responses were defined as ‘mutant-selective’ if there was significant reactivity against the mutant epitope but ‘some’ reactivity against the wild type epitope. This is not a clear definition, and some criteria should be used such as a 10-fold or greater difference between the response to the mutant and wild type peptide. It should also be pointed out that wild-type cross-reactive T cells will be ineffective at tumor control as these presumably represent targets that are not naturally processed and presented at levels on tumor cells.

We appreciate the reviewer’s request for additional clarity on this. The criteria we have implemented to define a mutant selective response is as defined below. This does not change the number of responses that fall into each category that was described previously:

“Clones that showed significant reactivity to the MT peptide over the unloaded DC control, and significant reactivity to the WT peptide compared to the unloaded DC control, additionally, a significant difference should be observed between MT and WT reactivities”

> **Change:** We have addressed this by updating the language in the following places. Main file: lines 323 – 324; lines 851 – 85. Supplemental file: lines 208 – 218.

Regarding the assertion that wildtype cross-reactive T cells are ineffective in tumor control, we propose an alternative perspective. Single point mutations can influence the processing and presentation of peptides in either direction, either reducing or enhancing their likelihood of being processed and presented. When these mutations lead to more efficient peptide presentation, they may also enhance T cell recognition¹. We have clarified this in the text.

> **Change (lines 214 – 216):** *“Given that neoantigen epitopes can differ from their wild-type (WT) counterparts by as little as a single amino acid, and single point mutations can either reduce or*

enhance the likelihood of the epitope to be processed and presented¹, we evaluated whether the induced T cell responses could differentiate between mutant (MT) and WT peptides.”

The response to question 7 from reviewer 2 was unclear. The statement that was added to the text indicated that six had stable disease as their best overall response but four of these patients showed tumor regression. Why was the best overall response of these four patients not designated tumor regression? Was tumor regression based upon RECIST criteria, and what was the duration of responses seen in these patients?

We thank the reviewer for their request for clarification. Indeed, RECIST 1.1 criteria were used to define the observed best overall responses (BOR). The observed tumor regression (up to -20% reduction in target lesions) did not meet criteria for partial response (which is defined as at least -30% decrease in the sum of target lesions, per RECIST 1.1). Therefore, the best overall response is defined as SD. The duration of the response of these four patients was between 12 and 36 weeks. We have updated the language in the discussion to incorporate these edits.

> Change (lines 445 – 448): *“Of the nine treated patients, six had stable disease (SD) as their best overall response, including up to 36 weeks for one patient. Four of these patients with SD showed tumor regression (up to -20% according to Response Evaluation Criteria in Solid Tumors (RECIST) version 1.1) after infusion.”*

Reviewer #2 (Remarks to the Author):

The revision and rebuttal are careful, specific, and, on the whole, responsive. Most of my major and minor concerns are fully or substantially addressed with added methods detail, clearer figure legends, new supplemental analyses (specificity categories; functional avidity), and expanded clinical context and discussion. Only two items remain partially addressed (noted below). Explicit clinical release criteria for WT-cross-reactive clones, and breadth of claims for low-TMB tumors. Otherwise, the manuscript is materially clearer and stronger.

1. The authors now summarize the distribution mutant-specific 7/19 (36.8%), mutant-selective 9/19 (47.4%), WT-cross-reactive 3/19 (15.8%) and provide full profiles in Suppl. Fig. 7, with example panels and a summary table in Fig. 4D. They also added functional avidity data for four TCRs in Suppl. Fig. 8, which was exactly the kind of orthogonal support I suggested. However, my request to clarify whether WT-reactive clones are excluded from clinical release is not answered explicitly; the rebuttal focuses on bulk-product infusion and post-hoc assessments. The new data are strong; a single sentence stating the release criterion will fully close this point.

We thank the reviewer for requesting the additional claim and we have included this in the manuscript.

> Change (lines 327 – 329): *“Given the infrequent observation of WT-reactive responses during development, the clinical approach² involved infusing bulk products without the exclusion of WT-reactive clones.”*

2. The new ovarian cancer data demonstrate feasibility beyond melanoma and are presented transparently; however, the reported TMB 25–88 spans moderate–high mutational loads, so it doesn't fully speak to the lowest TMB contexts I had in mind. Still, adding these data and updating counts strengthens generalizability.

We appreciate that the reviewer recognizes the value of the additional data generated in the context of OVC. For transparency purposes, we have included a reference to Alexandrov *et al.*³ that describes OVC and melanoma as indications with a moderate-to-high mutational load.

> **Change (lines 167 – 168):** “...we applied the optimized process to PBMCs from melanoma patients and ovarian cancer (OVC) patients, indications with a mid-to-high mutational load³.”

Reviewer #3 (Remarks to the Author):

The authors have provided a careful and detailed rebuttal, and several of my earlier points have been appropriately addressed. In particular, the revisions related to donor inclusion and diversity (Question 1), antigen immunogenicity classification (Question 6), and figure and marker clarity (Questions 7 and 8) improve the manuscript's transparency and readability. These efforts demonstrate the authors' commitment to strengthening the presentation and completeness of their work.

However, several core scientific concerns remain insufficiently resolved.

Regarding Question 2, the inclusion of tumor antigen expression and clonality data represents only a minimal and largely cosmetic fix. The new data are not analyzed or discussed, merely appended to a supplementary spreadsheet. There is no examination of how antigen expression correlates with immune responses, nor any commentary on tumor purity or clonal context. While this addition improves transparency, it adds little biological or clinical insight.

We appreciate the reviewer's request for additional insights from this data. We have not emphasized such data in our manuscript because we believe that the relationships we observe are more influenced by the design of NEO-STIM and the principles of our target selection framework than by inherent biological processes.

It is well documented that high antigen expression correlates with the ability to raise immune responses in patients⁴, our in vitro system does not display this correlation (see figure below). This is not unexpected as we add equimolar amounts of the target peptides for stimulation, rendering the natural gene expression level of the target gene irrelevant when raising T cell responses with NEO-STIM. This is important because our data shows that most of the responses we observe are induced *de novo* during manufacturing rather than expanding out pre-existing responses. Naturally, high expression of antigen is expected to remain relevant for anti-tumor reactivity upon infusion of the manufactured drug product.

Regarding clonality/variant allele fraction (VAF), it is important to note that we cannot assess the immunogenicity of all mutations/epitopes in our system, so we employ a target selection framework. Through the computational algorithm, we prioritize peptides based on primary characteristics like binding affinity, antigen expression, and proteasomal cleavage potential⁵, an approach that has been shown to allow for the discrimination of immunogenic peptides in a large

consortium as well⁴. We complement this by secondary characteristics such as VAF for refining epitope selection⁵. This approach emphasizes more clonal targets due to their clinical significance^{6,7} but also ensures that lower clonality targets with robust primary traits remain in the pool. Therefore, the observed phenomenon that immunogenic peptides exhibit a lower median VAF compared to non-responsive ones (see figure below) is likely a reflection of the selection process rather than intrinsic biological factors. Again, high VAF is expected to be relevant for anti-tumor reactivity and prevention of antigen escape upon infusion of the manufactured drug product.

For Question 3, the key issue of functional validation remains. The authors continue to rely primarily on engineered A375 tumor cells, which are artificial targets that do not fully reflect native tumor antigen processing or presentation. No new functional data were added, and the rebuttal mainly reinterprets existing results rather than addressing the underlying experimental limitation. Similarly, for Question 4, the justification provided for the use of artificial tumor models is pragmatic but not a solution. Citing another study from the group offers useful context but does not compensate for the lack of validation within this work. The connection is indirect and reads more as an appeal to external support than as additional evidence. For a mechanistic or preclinical study, this remains a significant shortfall.

We share the reviewer's commitment to using physiologically relevant models. In the context of a personalized neoantigen-specific product, the unique HLA background and mutational landscape of each patient mean that only autologous tumor material can truly reflect native tumor antigen processing and presentation. This manuscript includes recognition of autologous tumors from two patients for whom we had access to this type of material, 1 T cell response was tested from PT04, n = 3 replicates; and 2 T cell responses were tested for PT05, n = 3 replicates each). While we acknowledge that the A375 tumor cells used in our study require engineering of the HLA and the immunogen components, which do not fully capture the physiological situation in patients, we believe that they still offer valuable insights as presented in this manuscript. Furthermore, we remain confident that the reference to the clinical data generated with this manufacturing process supports the functional validation presented in this preclinical manuscript.

We have acknowledged the engineered nature of using A375 tumor cells in the context of tumor recognition:

> **Change (lines 409 – 414):** “Finally, the manufactured drug products (DPs) demonstrated cytotoxic potential against tumor cells engineered to express the patients’ specific HLA and neoantigens. Although these targets are engineered cell lines, we additionally observed recognition of autologous tumor material for two patients, further supporting the relevance of our findings.”

With respect to Question 5, the clarification regarding CD4⁺ T-cell baseline controls is appreciated, but no new data have been added. The CD4⁺ component remains underexplored. While the limitation of available multimer reagents is understandable, reviewers would generally expect alternative or creative approaches to strengthen this aspect in a high-impact submission.

We respectfully disagree that the CD4⁺ component remains underexplored. With the addition of the ovarian cancer data (Supplementary Figure 6), we strengthened the manuscript with an additional 18 identified CD4⁺ T cell responses. We further show functional validation that these induced CD4⁺ T cell responses also produce multiple effector functions (CD107a degranulation, TNF α , and IFN γ production).

Overall, while the revised manuscript is clearer and more complete in presentation, the experimental depth and validation required to fully support the central claims are still lacking.

- 1 Finnigan, J. P. *et al.* Structural basis for self-discrimination by neoantigen-specific TCRs. *Nat Commun* **15**, 2140 (2024). <https://doi.org/10.1038/s41467-024-46367-9>
- 2 Borgers, J. S. W. *et al.* Personalized, autologous neoantigen-specific T cell therapy in metastatic melanoma: a phase 1 trial. *Nat Med* **31**, 881-893 (2025). <https://doi.org/10.1038/s41591-024-03418-4>
- 3 Alexandrov, L. B. *et al.* Signatures of mutational processes in human cancer. *Nature* **500**, 415-421 (2013). <https://doi.org/10.1038/nature12477>
- 4 Wells, D. K. *et al.* Key Parameters of Tumor Epitope Immunogenicity Revealed Through a Consortium Approach Improve Neoantigen Prediction. *Cell* **183**, 818-834 e813 (2020). <https://doi.org/10.1016/j.cell.2020.09.015>
- 5 Ott, P. A. *et al.* A Phase Ib Trial of Personalized Neoantigen Therapy Plus Anti-PD-1 in Patients with Advanced Melanoma, Non-small Cell Lung Cancer, or Bladder Cancer. *Cell* **183**, 347-362 e324 (2020). <https://doi.org/10.1016/j.cell.2020.08.053>
- 6 McGranahan, N. *et al.* Clonal neoantigens elicit T cell immunoreactivity and sensitivity to immune checkpoint blockade. *Science* **351**, 1463-1469 (2016). <https://doi.org/10.1126/science.aaf1490>
- 7 Hoyos, D. & Greenbaum, B. D. Perfecting antigen prediction. *J Exp Med* **219** (2022). <https://doi.org/10.1084/jem.20220846>